# Distributional Active Inference

Abdullah Akgül [1]   Gulcin Baykal [1]   Manuel Haußmann [1]   Mustafa Mert Çelikok [1]   Melih Kandemir [1]

## Abstract

Optimal control of complex environments with robotic systems faces two complementary and intertwined challenges: efficient organization of sensory state information and far-sighted action planning. Because the reinforcement learning framework addresses only the latter, it tends to deliver sample-inefficient solutions. Active inference is the state-of-the-art process theory that explains how biological brains handle this dual problem. However, its applications to artificial intelligence have thus far been limited to extensions of existing model-based approaches. We present a formal abstraction of reinforcement learning algorithms that spans model-based, distributional, and model-free approaches. This abstraction seamlessly integrates active inference into the distributional reinforcement learning framework, making its performance advantages accessible without transition dynamics modeling.

## 1. Introduction

The human brain is an experience machine (Clark, 2024): a survival system that enables far-sighted planning by efficiently organizing multimodal sensory input. As the most advanced general intelligence we know, it is a inspiration for autonomous agents. Yet it remains unclear how the brain structures sensory stimuli into decision-relevant variables to solve control problems efficiently (Dorrell et al., 2023). Reinforcement learning (RL) formalizes far-sighted planning, and recent advances enable web-scale learning of multimodal distributions for general-purpose action generation (Kim et al., 2024; Black et al., 2024; NVIDIA Research et al., 2025). Still, we lack an account of how such representations are organized for efficient planning under computational and data constraints (Botteghi et al., 2025).

The *active inference framework (AIF)* offers a process-level theory of whole-brain information organization (Friston et al., 2017; Parr et al., 2022). It posits an action–perception cycle in which neural dynamics for perception and control evolve to minimize a single objective, the expected free energy (EFE) (Friston et al., 2015). Formally, this amounts to variational Bayes on a controlled Markov process, yielding a form of model-predictive control. Its links to modern machine learning have motivated AIF-inspired adaptive control methods (Millidge et al., 2020; Tschantz et al., 2020b; Millidge, 2021; Lanillos et al., 2021; Schneider et al., 2022; Malekzadeh & Plataniotis, 2024), but in practice these efforts have largely rederived familiar information-theoretic exploration heuristics (e.g., information gain) (Houthooft et al., 2016; Sukhija et al., 2025). To date, AIF has not delivered major practical gains in state-of-the-art RL.

Observing that the brain is an efficient information-processing system, we investigate whether AIF is useful in situations where high-fidelity forward simulation is infeasible. *Distributional RL* (Bellemare et al., 2017; 2023) captures the full return distribution, which encodes information about future state transitions that would otherwise require an explicit forward model. This makes distributional RL a natural vehicle for incorporating AIF without the cost of learning an explicit transition model, which we pursue here. We achieve this goal in three steps. Firstly, we analyze AIF strictly through the lens of variational Bayesian and causal inference, reconstructing its formulation by applying a mechanistic deduction chain over a coherent set of formal entities. By expressing the prior construction via do-calculus (Pearl, 1995), we show that the standard objective admits a simpler equivalent of its commonly adopted form.

Secondly, we formalize a new theoretical framework, called *push-forward RL*, that explains the return distribution as pushing the trajectory measure of a specific policy-induced state transition kernel forward with a return functional. This recipe enables us to relate Bellman updates to the transition kernels implied by the resulting push-forwards of trajectory measures, thereby connecting model-based and model-free views of policy iteration. We use this connection to embed active inference into the distributional RL framework.

Lastly, we exploit our theory to design a general-purpose policy optimization algorithm called *Distributional Active Inference (DAIF)*. DAIF is an intuitive and easily implementable extension of distributional RL. Figure 1 provides

[1]University of Southern Denmark. Correspondence to: Melih Kandemir <kandemir@imada.sdu.dk>.

*Proceedings of the 43rd International Conference on Machine Learning*, Seoul, South Korea. PMLR 306, 2026. Copyright 2026 by the author(s).

a conceptual overview of the framework. DAIF performs temporal-difference quantile matching on a probabilistic embedding space constructed by a state-action amortized parametric distribution. Across a broad suite of tabular and continuous-control tasks, this simple modification delivers substantial performance gains. The results support our view that AIF is particularly powerful when the agent has limited computational capabilities, mirroring the conditions of the biological brains it is intended to explain.

**Notation.** We denote probability measures by $P$, and by $\mathcal{P}_\mathcal{X}$ the set of all probability measures on the measurable space $(\mathcal{X}, \sigma(\mathcal{X}))$ for a sigma-algebra $\sigma(\mathcal{X})$. Marginalization of a random variable and conditioning on a point observation $z$ are denoted as

$$P_{X|Y}P_{Y|z} \triangleq \int P_{X|Y}(\cdot|Y)P_{Y|Z}(dY|z) = P_{X|z}.$$

We suppress the conditioning on variables that are integrated out, e.g., $P_X P_Y$ instead of $P_{X|Y}P_Y$. By $F_P$ and $f_P$ we denote the cumulative distribution function and probability density function of probability measure $P$, respectively. The $p$-Wasserstein distance between two measures $P, \bar{P}$ is

$$\mathcal{W}_p(P, \bar{P}) \triangleq \big( \inf_{\nu \in \Gamma(P, \bar{P})} \mathbb{E}_{(x,x')\sim\nu}[|x - x'|^p] \big)^{1/p},$$

where $\Gamma(P, \bar{P})$ denotes the set of couplings. We write $\delta_x$ for the Dirac measure at $x$, defined by $\delta_x(A) = \mathbf{1}(x \in A)$ for measurable set $A$. The $\mathrm{do}(X \sim P_{\mathrm{new}})$ operator denotes a stochastic intervention (Pearl, 1995) that replaces the distribution of variable $X$ in a structural causal model by $P_{\mathrm{new}}$. See Table 2 in the appendix for the full notation.

**Problem Statement.** We consider controlled Markov processes defined by a transition kernel $P(X'|X, A)$ where $X, X' \in \mathcal{X}$ denote the current and next environment states, and $A \in \mathcal{A}$ indicates the action taken by the agent, with an unknown transition kernel $P_*$. The agent has a reward function $R : \mathcal{X} \times \mathcal{A} \to [0, R_{\max}]$, a *world model* $P_W(X, S) : \mathcal{X} \times \mathcal{S} \to [0, 1]$ that defines a joint distribution over environment states and auxiliary variables $S$ in a latent embedding space $\mathcal{S}$, and a policy $\pi : \mathcal{X} \to \mathcal{A}$. The agent seeks to maximize its adaptation to the environment by updating $P_W$ and $\pi$ through its interactions with the environment which evolves according to $P_\pi^*$. The agent updates its policy $\pi$ by estimating the discounted sum of rewards, called the *return functional*, after taking action $a_0$ in state $x_0$ and following a fixed policy $\pi$:

$$\mathbf{G}_{x_0,a_0}^\pi(\omega) \triangleq R(x_0, a_0) + \sum_{t=1}^\infty \gamma^t R(x_t, \pi(x_t))$$

for some discount factor $\gamma \in (0, 1)$ and a trajectory $\omega \triangleq (x_1, x_2, \dots) \in \mathcal{X}^{\mathbb{N}_+}$.

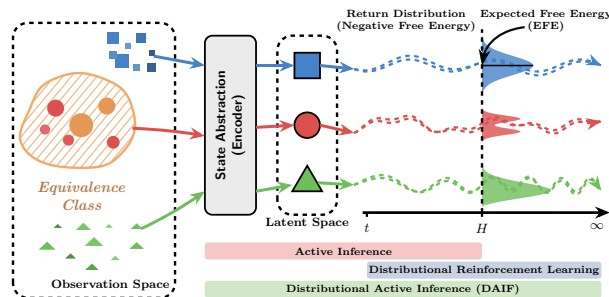

*Figure 1.* Conceptual overview of the distributional active inference (DAIF) framework. Similar states are grouped into abstract states on a latent state space via a probabilistic encoder. Infinite-horizon planning is performed on the latent state space using *distributional reinforcement learning*. The objective function of the resulting pipeline is derived from the principles of *active inference*.

## 2. A Rigorous Formulation of Active Inference

Active inference has substantial empirical support as a meta-level process theory for cognitive neuroscience (Hodson et al., 2024). However, its algorithmic details and probability-theoretic implications allow diverse interpretations. We will next derive the AIF training objective from the first principles of Bayesian and causal inference. The outcome yields important simplifications, which paves the way for the integration of AIF into distributional RL.

Let the agent's world model be

$$P_W(X, Y, S) \triangleq P_D(X|Y, S)P_0(Y, S),$$

with a prior belief over the latent embeddings $P_0(Y, S)$ and a decoding likelihood $P_D(X|Y, S)$. Variable $Y$ represents actions and variable $S$ represents perceptions. The Evidence Lower Bound (ELBO) functional for observation $x$ and approximate posterior $P_Q(Y, S)$ follows from the standard machinery of variational inference

$$L(x, P_W, P_Q) = \\ \mathbb{E}_{y,s\sim P_Q}\Big[ \log\big( P_D(x|y, s)P_0(y, s)/P_Q(y, s)\big)\Big].$$

Maximizing this functional with respect to $Q$ minimizes the Kullback-Leibler divergence $\mathrm{KL}(P_Q(Y, S)||P(Y, S|x))$, as this quantity equals the Jensen gap between $\log P(X)$ and $L(x, P_W, P_Q)$. AIF is defined as an extension of variational inference to control problems in which the transition kernel and the policy are jointly optimized with the same ELBO. AIF aims to solve $P_Q := \arg\max_{\bar{P}_Q} L(x, P_W, \bar{P}_Q)$ with

$$f_{P_0}(Y := y) \propto \exp(\mathbb{E}_{x'\sim P_{W|\mathrm{do}(Y\sim\delta_y, S\sim P_Q)}}[ \\ L(x', P_{W|\mathrm{do}(Y\sim\delta_y, S\sim P_Q)}, P_Q)])$$

where the intervention $\mathrm{do}(S \sim P_Q)$ is intended to incorporate the effect of past observations on future predictions. Up

to a sign, the exponent above coincides with the expected free energy (EFE) of conventional active inference (Friston et al., 2015); we adopt the ELBO formulation here for closer alignment with the variational-inference machinery used in the rest of the paper. A model equipped with such a prior would favor a world model whose predicted observations best fit it. This principle is known in cognitive neuroscience as *predictive coding* (Rao & Ballard, 1999; Friston, 2009). AIF implements a particular way of predictive coding by intervening into the world model used in the ELBO with a *desired state distribution* $P_R(X)$, which plays the role of the reward function in RL. The canonical AIF prior construction applies the product rule over the approximate world model in the reverse direction

$$P_{W|\text{do}(Y,S\sim P_Q)}(X,Y,S) = P_D(X|Y,S)P_Q(Y,S)$$
$$= \widetilde{P}(Y,S|X)\widetilde{P}(X)$$

which yields an approximate posterior $\widetilde{P}(Y,S|X)$ and the corresponding prior $\widetilde{P}(X)$. The prior $\widetilde{P}(X)$ is then intervened by $P_R$. We now point to an overlooked matter in the AIF literature. Due to the equality imposed by the product rule, the intervention on $X$ yields (see Appendix A.4):

$$P_{\widetilde{W}}(X,Y,S) \triangleq \tag{1}$$
$$P_{W|\text{do}(Y,S\sim P_Q,X\sim P_R)}(X,Y,S) = P_Q(Y,S)P_R(X)$$

and disconnects the observable $X$ from the latents $(Y,S)$. The key consequence is that the intervened distribution of $(Y,S)$ becomes independent from $X$, eliminating the effort discussed in prior work about the inference of $\widetilde{P}(Y,S|X)$ (Da Costa et al., 2020). Substituting this outcome into the prior construction developed above, we obtain

$$f_{P_0}(Y := y) \propto \exp(\mathbb{E}_{x'\sim P_D P_{Q|y}}[\log P_R(x')])$$

which yields the complete AIF objective below

$$L(x, P_W, P_Q) =$$
$$\mathbb{E}_{y,s\sim P_Q}[\log P_D(x|y,s)] + \mathbb{E}_{y,s\sim P_Q}[\log P_0(s|y)] \tag{2}$$
$$+ \mathbb{E}_{x'\sim P_D P_Q}[\log P_R(x')] + \mathbb{H}[P_Q].$$

See Appendix A.5 for the intermediate derivation steps.

Next, we connect the minimalist derivation presented above to the control setting. We exploit the factorization in Eq. 1 to simplify the ELBO further. The world model of an AIF agent comprises a policy variable $\pi$ to represent actions and the latent embedding $(S, S')$ that represents the current perceptions by $S$ and the future perceptions by $S'$. AIF theory assumes that these variables factorize as:

$$P_W(X', S', S, \pi|X, A) \triangleq$$
$$P_D(X'|S')P_M(S'|S,A)P_E(S|X)P_A(\pi)$$

where $P_M$ is a latent state transition kernel, $P_E : \mathcal{X} \to \mathcal{P}_S$ is an encoder that maps observations to latent perceptions, $P_D : \mathcal{S} \to \mathcal{P}_{\mathcal{X}}$ is a decoder that maps in the reverse direction, and $P_A$ is a policy distribution. Note that the generic action and perception variables $(Y, S)$ from the preceding derivation are specialized here to the control-specific $(\pi, (S, S'))$; subsequent occurrences of $P_W$ and $P_Q$ in this section refine—rather than redefine—these earlier objects. Let us assume an approximate posterior

$$P_Q(S, S', \pi) \triangleq P_M(S'|S, A)P_E(S|X)P_A(\pi|X)$$

that shares the same transition kernel and encoder with the world model. Such a choice of approximate posterior is standard in the probabilistic state-space modeling literature (Doerr et al., 2018) and is used to eliminate the latent overshooting term in the Dreamer model family (Hafner et al., 2020a; 2025). With this approximate posterior, the term $\mathbb{E}_{y,s\sim P_Q}[\log P_0(s|y)]$ developed in Eq. 2 drops and we get:

$$L(x, a, x', P_W, P_Q) =$$
$$\mathbb{E}_{S'\sim P_M P_E|x,a}[\log P_D(x'|S')]$$
$$+ \mathbb{E}_{x'',\pi\sim P_D P_M P_E P_A|x'}\Big[\log P_R(x''|x', \pi(x'))\Big] \tag{3}$$
$$+ \mathbb{H}[P_{A|x'}].$$

In RL terms, this ELBO prescribes a Dyna-style model-based approach. The first term learns the state transition dynamics from observed triples $(x, a, x')$. The remaining two terms can be viewed as maximum-entropy policy search (Haarnoja et al., 2018a;b) on the estimated transition model. The formulation can be straightforwardly extended to the case where $x''$ is an arbitrarily long trajectory.

## 3. Push-forward Reinforcement Learning

We project AIF onto the distributional RL framework via a model-based extension of policy iteration presented in Algorithm 1, which augments standard policy iteration with a model update step. It performs policy updates by Bellman backups using the following operator, where the backup transition kernel $P$ comes from the estimated model

$$(T_P^\pi Q)(x, a) \triangleq$$
$$R(x, a) + \gamma \mathbb{E}_{x'\sim P(\cdot|x,a)}[Q(x', \pi(x'))] \tag{4}$$

where $Q : \mathcal{X} \times \mathcal{A} \to \mathbb{R}$ is an arbitrary function. $\mathcal{U}(\mathcal{X})$ in Algorithm 1 is an idealized full-coverage assumption; $x$ is drawn from the state distribution induced by the data (e.g., under a behavior policy). MBPI is best seen as a template that highlights design choices shared by modern model-based RL. For instance, PSRL samples an MDP from the posterior and solves it (Osband et al., 2013); PILCO combines model learning with policy search (Deisenroth & Rasmussen, 2011); and Dreamer couples model learning

---

**Algorithm 1** Model-Based Policy Iteration (MBPI)

**Input:** $\pi_0, i := 0$
**while** True **do**

    Model Update:

$$P_{i+1} := \arg\max_P \{$$
$$\mathbb{E}_{x \sim \mathcal{U}(\mathcal{X})} \mathbb{E}_{x' \sim P_*(\cdot|x, \pi_i(x))}[\log f_P(x'|x, \pi_i(x))]\}$$

    Policy Evaluation:

$$Q_{i+1} := \arg\min_Q \mathbb{E}_{x \sim \mathcal{U}(\mathcal{X})} \Big[\big(T^{\pi_i}_{P_{i+1}} Q(x, \pi_i(x))$$
$$- Q(x, \pi_i(x))\big)^2\Big]$$

    Policy Improvement:

$$\pi_{i+1} := \arg\max_\pi \mathbb{E}_{x \sim \mathcal{U}(\mathcal{X})} \Big[Q_{i+1}(x, \pi(x))\Big]$$
$$i := i + 1$$

**end while**

---

with actor–critic updates (Hafner et al., 2020a). We suppress the policy entropy maximization in Eq. 3 for brevity. With finite samples and finite backups, the algorithm inherits the properties of Approximate Policy Iteration (API), a two-error scheme (Bertsekas & Tsitsiklis, 1996, Prop. 6.2): each iteration incurs (i) a policy-evaluation error $\varepsilon$ and (ii) an approximate greedification error $\delta$, yielding asymptotic suboptimality of order $(\delta + 2\gamma\varepsilon)/(1 - \gamma)^2$. In API, $\varepsilon$ and $\delta$ are controlled from data; worst-case guarantees such as $|Q(x, a) - \mathbb{E}[\mathbf{G}^\pi_{x,a}]|_\infty \leq \varepsilon$ demand accurate estimates over a large set of states, so the total sampling burden is dominated by the size of that set. For high-dimensional state spaces, $|\mathcal{X}|$ can grow rapidly with dimension under discretization, motivating learned state abstractions that group states with similar transition dynamics in a latent embedding (Li et al., 2006; Abel et al., 2016; 2018).

As the second step, we generalize the distributional RL framework to the case in which the transition dynamics are explicitly identified as an intermediate step and then pushed forward on the trajectory measures. Approaching RL from a probabilistic perspective, this decomposition will pave the way to integrate AIF into the distributional RL framework. Since the return functional $\mathbf{G}^\pi_{x,a}(\omega)$ operates on a state-action trajectory $\omega$, its distribution (therefore expectation) can be expressed through a probability measure over trajectories. We construct such a measure below.

Let $P^\pi(B|x) \triangleq P(B|x, \pi(x))$ be the Markov kernel induced by a policy $\pi : \mathcal{X} \to \mathcal{A}$ for $B \in \sigma(\mathcal{X})$. By the Ionescu-Tulcea theorem (Ionescu-Tulcea, 1949), the iterated application of $P(\cdot|x_0, a_0)$ followed by $P^\pi$ extends uniquely to a probability measure $\mathbb{P}^{P^\pi}_{x_0, a_0}$ on the path space $(\mathcal{X}^{\mathbb{N}_+}, \sigma(\mathcal{X})^{\otimes \mathbb{N}_+})$ that is consistent with the finite-dimensional kernel products on every cylinder set; see Appendix A.7 for the explicit construction. We refer to objects like $\mathbb{P}^{P^\pi}_{x_0, a_0}$ as *Markov process measures*.

Let $(E, \sigma(E))$ be a measurable space and $f : \mathcal{X} \to E$

be a measurable map. We extend this to the infinite-horizon path space by defining the sequence-valued map $\mathbf{F} : \mathcal{X}^{\mathbb{N}_+} \to E^{\mathbb{N}_+}$ as $\mathbf{F}(\omega) = (f(x_1), f(x_2), \dots)$ where $\omega = (x_1, x_2, \dots)$ is an element of $\mathcal{X}^{\mathbb{N}_+}$. We refer to objects like $\mathbf{F}$ as *push-forward process functionals*. Pushing a Markov measure forward with this functional, we get a measure on the observation path space $(E^{\mathbb{N}_+}, \sigma(E)^{\otimes \mathbb{N}_+})$:

$$\mathbf{F}_\# \mathbb{P}^{P_\pi}_{x_0, a_0} \triangleq \mathbb{P}^{P_\pi}_{x_0, a_0}(\mathbf{F}^{-1}(B))$$
$$= \mathbb{P}^{P_\pi}_{x_0, a_0}\{\omega \in \mathcal{X}^{\mathbb{N}_+} : \mathbf{F}(\omega) \in B\}$$

for any measurable set $B \in \sigma(E)^{\otimes \mathbb{N}_+}$. Under this definition, the measure $(\mathbf{G}^\pi_{x_0, a_0})_\# \mathbb{P}^{P_\pi}_{x_0, a_0}$ corresponds precisely to the *return distribution* on the measurable space $(\mathbb{R}, \mathcal{B}(\mathbb{R}))$. For any Borel set $B \in \mathcal{B}(\mathbb{R})$, this measure can be expressed as:

$$\mathbb{P}^{P_\pi}_{x_0, a_0}\Big(\Big\{\omega \in \mathcal{X}^{\mathbb{N}_+} : $$
$$R(x_0, a_0) + \sum_{t=1}^\infty \gamma^t R(x_t, \pi(x_t)) \in B\Big\}\Big).$$

The return distributions induced by two different Markov kernels $P$ and $P'$ can then be expressed as $(\mathbf{G}^\pi_{x_0, a_0})_\# \mathbb{P}^{P_\pi}_{x_0, a_0}$ and $(\mathbf{G}^\pi_{x_0, a_0})_\# \mathbb{P}^{P'_\pi}_{x_0, a_0}$, respectively.

We define the distributional counterpart of the generalized Bellman operator introduced in Eq. 4 with a specific Markov kernel $P$ as follows:

$$\mathbf{T}^\pi_P \eta_{x,a} \triangleq R(x, a) + \gamma \mathbb{E}_{x' \sim P(\cdot|x,a)}[\eta_{x', \pi(x')}]$$

for a real-valued probability measure $\eta_{x,a} : \mathcal{X} \times \mathcal{A} \to \mathcal{P}_\mathbb{R}$ conditioned on $(x, a) \in \mathcal{X} \times \mathcal{A}$. Define the maximal form of the $p$-Wasserstein distance $\bar{\mathcal{W}}_p(P, \bar{P}) \triangleq \sup_{x,a} \mathcal{W}_p(P_{x,a}, \bar{P}_{x,a})$ where $P_{x,a}$ and $\bar{P}_{x,a}$ are two probability measures on the same space as $\eta_{x,a}$. We can re-express the contraction property of the distributional Bellman operator (Bellemare et al., 2017) in terms of push-forwards of Markov process measures.

**Lemma 3.1.** *Let $P_\pi, \bar{P}_\pi, P^*_\pi$ be Markov kernels induced by a fixed policy $\pi$ and $\mathbf{F}$ be a push-forward process functional, then the distributional Bellman operator $\mathbf{T}^\pi_{P_*}$ is a contraction with respect to $\bar{\mathcal{W}}_p$*

$$\bar{\mathcal{W}}_p\Big(\mathbf{T}^\pi_{P_*} \mathbf{F}_\# \mathbb{P}^{P_\pi}, \mathbf{T}^\pi_{P_*} \mathbf{F}_\# \mathbb{P}^{\bar{P}_\pi}\Big) \leq \gamma \bar{\mathcal{W}}_p\Big(\mathbf{F}_\# \mathbb{P}^{P_\pi}, \mathbf{F}_\# \mathbb{P}^{\bar{P}_\pi}\Big).$$

The key nuance is that the contraction modulus is due to $\gamma$ within the distributional Bellman operator, not to the discount factor applied by the return functional. From Banach's fixed point theorem (Banach, 1922), it follows that there exists $\widehat{P}_\pi \in \mathcal{P}_\mathcal{X}$ that satisfies $\bar{\mathcal{W}}_p(\mathbf{T}^\pi_{P_*} \mathbf{F}_\# \mathbb{P}^{\widehat{P}_\pi}, \mathbf{F}_\# \mathbb{P}^{\bar{P}_\pi}) = 0$ for a sufficiently large set of admissible Markov kernels $\mathcal{P}_\mathcal{X}$. One can find this fixed point at a geometric rate (i.e., the error is $O(\gamma^k)$) by starting from an arbitrary $P^0_\pi \in \mathcal{P}_\mathcal{X}$ and

repeatedly applying $\mathbf{T}_{P_*}^\pi$. This fixed point is unique in the space of Markov process measures after a push-forward with functional $\mathbf{F}$, as guaranteed by the Banach fixed point theorem applied to the contraction established in Lemma 3.1: since $\gamma < 1$, $\mathbf{T}_{P_*}^\pi$ is a strict contraction on the complete metric space $(\mathcal{P}_{\mathbb{R}}^{\mathcal{X} \times \mathcal{A}}, \bar{\mathcal{W}}_p)$, which admits exactly one fixed point. There may be multiple Markov process measures almost surely equal to this fixed point $\mathbf{F}_\# \mathbb{P}_{x_0,a_0}^{\hat{P}_\pi}$ after being pushed forward with $\mathbf{F}$. Evidently, $\mathbf{F}_\# \mathbb{P}_{x_0,a_0}^{P_\pi^*}$ also satisfies this condition. The Markov kernel of the Bellman operator anchors the search process performed via Bellman backups.

The advantage of distributional RL over model-based RL appears in situations where capturing the push-forward of the Markov process measure with the return functional is sufficient for decision making, such as in risk-sensitive RL (Lim & Malik, 2022; Keramati et al., 2020; Dabney et al., 2018a; Bernhard et al., 2019). The equivalence of this measure within a set of transition kernels brings sample efficiency when it can be exploited by the learning algorithm. Model-based RL theory formulates such equivalence classes via *state abstractions* (Li et al., 2006; Givan et al., 2003; Jiang et al., 2015). We will next develop some essential concepts to establish the link between state abstractions and distributional RL. First, we need to construct an embedding space on which state abstractions can be formulated.

**Definition 3.2.** *A mapping $K : \mathcal{S} \to \mathcal{P}_\mathcal{X}$ is said to be an $L$-**Lipschitz Markov kernel** if (i) for every $B \in \sigma(\mathcal{X})$, $s \mapsto K(s, B)$ is measurable and (ii) for all $s_1, s_2 \in \mathcal{S}$: $\mathcal{W}_p(K(s_1, \cdot), K(s_2, \cdot)) \leq L \cdot |s_1 - s_2|$. The action of the kernel $K$ on a measure $P \in \mathcal{P}_\mathcal{S}$, denoted $KP$, is the measure on $\mathcal{X}$ defined by:*

$$(KP)(B) \triangleq \int_\mathcal{S} K(s, B)\, P(ds), \quad \forall B \in \sigma(\mathcal{X}).$$

Now consider the push-forward of the transition kernel $P$ of a Markov process $\mathbb{P}_{x,a}^P$ with an $L_E$-Lipschitz continuous function $S$ to an embedding space $\mathcal{S}$, which we denote by $S_\# P$, and then mapping back to the state space by an $L_D$-Lipschitz Markov kernel $P_D$. This composite operation can be viewed as the action of $P_D$ on the measure $S_\# P$. We can also construct the same outcome by transforming the input of $P_D$, which then defines an operator on $P$.

**Definition 3.3.** *Let $F : \mathcal{X} \to \mathcal{S}$ be a measurable transformation, and $K : \mathcal{S} \to \mathcal{P}_\mathcal{X}$ a $L$-Lipschitz Markov kernel. We define the **composite kernel operator** $(KF) : \mathcal{X} \to \mathcal{P}_\mathcal{X}$ pointwise as $K(x, \cdot) = K(F(x), \cdot)$. For any measure $P \in \mathcal{P}_\mathcal{X}$, the action of the operator $KF$ is given by:*

$$(KF\, P)(B) \triangleq \int_\mathcal{X} K(F(x), B)\, P(dx), \quad \forall B \in \sigma(\mathcal{X}).$$

The observation is that $(KF\, P)$ acts identically to $K(F_\# P)$.

The following result establishes how an input transformation affects the Lipschitz continuity of a Markov kernel.

**Lemma 3.4.** *Let $F : \mathcal{X} \to \mathcal{S}$ be an $M$-Lipschitz transformation and $K : \mathcal{S} \to \mathcal{P}_\mathcal{X}$ be an $L$-Lipschitz Markov kernel. Then the composite kernel operator $(KF) : \mathcal{X} \to \mathcal{P}_\mathcal{X}$ is an $(L \cdot M)$-Lipschitz Markov kernel.*

Our main theoretical result follows from the fact that transforming the transition kernels of two Markov processes with a point-wise fixed $L$-Lipschitz operator contracts their probability measures by a factor of $L$ with respect to the $p$-Wasserstein distance. Consequently, performing distributional RL after an auto-encoding operation with an $L_E$-Lipschitz encoder $S_\# P$ and an $L_D$-Lipschitz decoder $P_D$ affects the contraction modulus by $L_E \cdot L_D$.

**Theorem 3.5.** *Let $P_\pi, \bar{P}_\pi, P_\pi^* \in \mathcal{P}_\mathcal{X}$ and $\mathbf{F}$ be a push-forward process functional. Let $P_D$ be an $L_D$-Lipschitz Markov kernel and $S : \mathcal{X} \to \mathcal{S}$ be a $L_E$-Lipschitz continuous measurable map from $(\mathcal{X}, \sigma(\mathcal{X}))$ to $(\mathcal{S}, \sigma(\mathcal{S}))$. Then the following inequality holds for any $x_0, a_0 \in \mathcal{A}, \mathcal{X}$*

$$\bar{\mathcal{W}}_p \left( \mathbf{T}_{P_*}^\pi \mathbf{F}_\# \mathbb{P}_{x_0,a_0}^{(P_D S)P_\pi}, \mathbf{T}_{P_*}^\pi \mathbf{F}_\# \mathbb{P}_{x_0,a_0}^{(P_D S)\bar{P}_\pi} \right)$$
$$\leq \gamma \cdot L_E \cdot L_D \cdot \bar{\mathcal{W}}_p \left( \mathbf{F}_\# \mathbb{P}_{x_0,a_0}^{P_\pi}, \mathbf{F}_\# \mathbb{P}_{x_0,a_0}^{\bar{P}_\pi} \right).$$

When a narrow information bottleneck yields efficient compression in the embedding space, $L_E$ should be small as the returns create large equivalence classes in the latent trajectory space (see Figure 4), which reduce the contraction modulus of the Bellman backups and speed up convergence. The decoder Lipschitz constant $L_D$ is small when the latent space captures the reward-relevant structure of the environment: nearby latent states map to nearby observations in Wasserstein sense, so $P_D$ need not stretch the metric. The product $L_E \cdot L_D$ formalizes the trade-off between compression and reconstruction fidelity—if the bottleneck is too narrow ($L_E$ very small), the decoder must compensate with a large $L_D$, potentially negating the convergence benefit.

We let distributional RL inherit these benefits of AIF by expressing auto-encoding within that framework. Let us redefine $\mathbf{G}_\# \mathbb{P}_{x_0,a_0}^{(P_D S)P_\pi}$ in terms of an *encoding process* defined on the latent space and a *decoding process* that maps back to the observation space. The encoding process on $\mathcal{S}^{\mathbb{N}_+}$ pushes $P_{x_0,a_0}^{P_\pi}$ forward through the sequence map $\mathbf{S}(\omega) = (S(x_1), S(x_2), \dots)$, which leads to a new measure $\mathbf{S}_\# \mathbb{P}_{x_0,a_0}^{P_\pi}$ on the space $(\mathcal{S}^{\mathbb{N}_+}, \sigma(\mathcal{S})^{\otimes \mathbb{N}_+})$. For any sequence $\mathbf{s} = (s_1, s_2, \dots) \in \mathcal{S}^{\mathbb{N}_+}$, the kernel $P_D$ induces a conditional path measure $\mathbb{P}^{P_D | \mathbf{s}}$ on $\mathcal{X}^{\mathbb{N}_+}$ via Ionescu-Tulcea:

$$\mathbb{P}^{P_D | \mathbf{s}}(C) \triangleq \int_{B_1} P_D(dx_1 | s_1) \times \cdots \times \int_{B_n} P_D(dx_n | s_n).$$

This represents the stochastic *decoding* of the representation path back into the state space. The return distribution can

now be expressed as an integration over the intermediate measure $\mathbf{S}_\# \mathbb{P}^{P_\pi}_{x_0,a_0}$ in the space $\mathcal{S}^{\mathbb{N}_+}$. Defining the expected return of a fixed representation path $\mathbf{s}$ as

$$(\mathbf{G}^\pi_{x_0,a_0})_\# \mathbb{P}^{P_D|\mathbf{s}}(B) \triangleq \mathbb{P}^{P_D|\mathbf{s}}\Big($$

$$\Big\{ \omega \in \mathcal{X}^{\mathbb{N}_+} : R(x_0,a_0) + \sum_{t=1}^\infty \gamma^t R(x_t, \pi(x_t)) \in B \Big\} \Big)$$

we can express the final return distribution as the integral of these local returns over the representation measure:

$$(\mathbf{G}^\pi_{x_0,a_0})_\# \mathbb{P}^{(P_D S)P_\pi}_{x_0,a_0}(B) = \int_{\mathcal{S}^{\mathbb{N}_+}} (\mathbf{G}^\pi_{x_0,a_0})_\# \mathbb{P}^{P_D|\mathbf{s}}(B)\, \mathbf{S}_\# \mathbb{P}^{P_\pi}_{x_0,a_0}(d\mathbf{s}). \tag{5}$$

The fact that the outcome is a composite measure will be instrumental in its implementation.

Let us next demonstrate how this result translates to a distributional RL algorithm. Although our result has more general implications, we will follow the established framework that performs Bellman residual minimization with respect to $\mathcal{W}_2$ for policy evaluation and a policy update on the expectation of the fitted return distribution. Formally, the policy evaluation step minimizes the objective below

$$J(P_\pi) \triangleq \mathbb{E}_{x \sim \mathcal{U}(\mathcal{X})}\big[\mathcal{W}_2^2((\mathbf{G}^\pi_{x_0,a_0})_\# \mathbb{P}^{P_\pi}_{x_0,a_0}, \mathbf{T}^\pi_{P_*}(\mathbf{G}^\pi_{x',\pi(x')})_\# \mathbb{P}^{P_\pi}_{x_0,a_0})\big].$$

Since $\mathbf{T}^\pi_{P_*}$ has a unique fixed point, we have $J(\widehat{P}_\pi) = 0$. When the Markov process is transformed by an auto-encoder $P_D S$ we attain the following upper bound to this objective

$$J(S) \triangleq \mathbb{E}_{x \sim \mathcal{U}(\mathcal{X})}\big[\mathcal{W}_2^2((\mathbf{G}^\pi_{x_0,a_0})_\# \mathbb{P}^{(P_D S)P_\pi}_{x_0,a_0}, \mathbf{T}^\pi_{P_*}(\mathbf{G}^\pi_{x',\pi(x')})_\# \mathbb{P}^{(P_D S)P_\pi}_{x_0,a_0})\big]$$

$$\leq \mathbb{E}_{x \sim \mathcal{U}(\mathcal{X})}\Big[\mathbb{E}_{\mathbf{s} \sim \mathbf{S}_\# \mathbb{P}^{P_\pi}_{x_0,a_0}}\Big[\mathbb{E}_{\tau \in \mathcal{U}(0,1)}\Big[\Big(F^{-1}_{(\mathbf{G}^\pi_{x_0,a_0})_\# \mathbb{P}^{P_D|\mathbf{s}}}(\tau) - F^{-1}_{\mathbf{T}^\pi_{P_*}(\mathbf{G}^\pi_{x',\pi(x')})_\# \mathbb{P}^{P_D|\mathbf{s}}}(\tau)\Big)^2\Big]\Big]\Big].$$

As the final statement is indexed only by $\mathbf{S}_\# \mathbb{P}^{P_\pi}_{x_0,a_0}$, we can instead search directly the space of encoding measures

$$\mathcal{Z}^\pi_{x,a} \triangleq \{Z_{x,a} : Z_{x,a} = \mathbf{S}_\# \mathbb{P}^{P_\pi}_{x_0,a_0}, P_\pi \in \mathcal{P}_\pi\} \tag{6}$$

for $(x,a) \in \mathcal{X} \times \mathcal{A}$. Using the expectation of the identified push-forward distributions in the policy improvement step—which is Bayes-optimal under squared loss, as the posterior mean minimizes the expected squared prediction error (see, e.g., Berger, 1985, Ch. 4)—we arrive at Algorithm 2. We call this algorithm template *push-forward policy iteration* and its practical applications *push-forward reinforcement learning*. This algorithm will deliver a policy sequence

---

**Algorithm 2** Push-Forward Policy Iteration (PPI)

**Input:** $\pi_0, i := 0$
**while** True **do**
  **for** $x \in \mathcal{X}$ **do**
    $Z^{i+1}_{x,a} := \arg\min_{Z \in \mathcal{Z}^{\pi_i}_{x,a}} \mathbb{E}_{\tau \sim \mathcal{U}(0,1)} \mathbb{E}_{\mathbf{s} \sim Z_{x,a}}\Big[$
      $\Big(F^{-1}_{(\mathbf{G}^\pi_{x,a})_\# \mathbb{P}^{P_D|\mathbf{s}}}(\tau) - F^{-1}_{\mathbf{T}^{\pi_i}_{P_*}(\mathbf{G}^{\pi_i}_{x',\pi_i(x')})_\# \mathbb{P}^{P_D|\mathbf{s}}}(\tau)\Big)^2\Big]$
    $\pi_{i+1}(x) :=$
      $\arg\max_{a \in \mathcal{A}} \mathbb{E}_{\mathbf{s} \sim Z^{i+1}_{x,a}}\Big[(\mathbf{G}^\pi_{x,a})_\# \mathbb{P}^{P_D|\mathbf{s}}(\mathbf{s})\Big]$
    $i := i + 1$
  **end for**
**end while**

---

identical to Algorithm 1 whenever $(\mathbf{G}^\pi_{x_0,a_0})_\# \mathbb{P}^{P_\pi}_{x_0,a_0}$ is realizable under $\mathcal{Z}^\pi_{x,a}$ for all $x, a, \pi$, as this will make their policy improvement steps measurably identical. In practice, this realizability can be enforced with powerful function approximators. The gain in sample complexity depends on the trade-off between the capacity of these function approximators and the level of compression that the underlying transition dynamics permits via state abstractions.

## 4. Active Inference with Push-Forward RL

We next integrate AIF into the push-forward RL framework and remove the need to estimate the transition kernel of the underlying dynamical system. We will exploit the fact that in Eq. 3, $P_M$ and $P_E$ always appear together in a marginalization process. Let us denote their marginal by

$$P_{S|x,a}(S') \triangleq \mathbb{E}_{s \sim P_{E|x}}[P_M(S'|s,a)].$$

This distribution can be interpreted as a *transfer transition kernel* from a source Markov chain on $\mathcal{X}$ to a target Markov chain on $\mathcal{S}$ (Lazaric & Restelli, 2011). $P_S$ is a push-forward of $P_\pi$ with some function $S$.

Let us define the next state $S'$ as the infinitely-long trajectory of future latent states $\mathbf{s}$ simulated on the world model after taking action $a$ at state $x$ and following policy $\pi$ afterwards. Let $\omega$ be the corresponding states in the observation space. We can then construct the desired state distribution as the return of this trajectory: $P_R(\omega) \propto \exp((\mathbf{G}^\pi_{x,a})_\#(\omega))$. Placing this construction into the related term of the ELBO developed in Equation (3) we get

$$\mathbb{E}_{\omega \sim P_D P_M P_E|x,a}[(\mathbf{G}^\pi_{x,a})_\#(\omega)]$$
$$= \mathbb{E}_{\omega \sim \mathbb{P}^{(P_D P_M P_E)\pi}_{x,a}}[(\mathbf{G}^\pi_{x,a})_\#(\omega)]$$
$$= \mathbb{E}[(\mathbf{G}^\pi_{x,a})_\# \mathbb{P}^{(P_D P_M P_E)\pi}_{x,a}]$$
$$= \mathbb{E}_{\mathbf{s} \sim \mathbf{S}_\# \mathbb{P}^{P_\pi}_{x,a}}\Big[(\mathbf{G}^\pi_{x,a})_\# \mathbb{P}^{P_D|\mathbf{s}}\Big]$$
$$= \mathbb{E}_{\omega \sim P_D P_{S|x,a}}\Big[(\mathbf{G}^\pi_{x',\pi(x')})_\#(\omega)\Big]$$

where we omit the integral over $P_A$ for notational brevity. The first and the fourth equations follow from definitions, the second from the Law of the Unconscious Statistician, and the third from Eq. 5. By learning an encoding process as in Equation (6) and performing distributional RL on the latent trajectory space, we can capture $(\mathbf{G}^\pi_{x,a})_\# \mathbb{P}^{P_D|\mathbf{s}}$. Thus, we can inherit the benefits of learning a transition kernel as in Algorithm 1 to simulate trajectories. Furthermore, we can quantify the effects of downstream calculations on these simulations in an infinitely long horizon.

We implement this recipe by learning a state-action amortized encoder that maps to the latent space and performing quantile regression of returns on this space. To operate on a latent space, we need to cast the quantile regression problem as a maximum likelihood estimation instance and assign input-dependent priors to its parameters. The solution of the problem below for an i.i.d. sample of $Y$ gives an unbiased estimate of the $\tau$'th quantile of a conditional distribution $P(Y|X)$ with input $X$ and output $Y$:

$$\widehat{g}_\tau \triangleq \arg\min_{g_\tau} \mathbb{E}_{X,Y \sim P}[\ell_\tau(Y - g_\tau(X))] \qquad (7)$$

for a predictor $g_\tau$ where $\ell_\tau(u) \triangleq (|u| + (2\tau - 1)u)/2$ is the check loss. By extending the connection established by Yu & Moyeed (2001), we can frame $\widehat{g}_\tau$ as the maximum likelihood estimate (MLE) of the Asymmetric Laplace Distribution (ALD) (Koenker & Bassett Jr, 1978)

$$f_\tau(Y|X; g_\tau, \sigma_\tau) = \frac{\tau(1-\tau)}{\sigma_\tau} \exp\left( -\frac{\ell_\tau(Y - g_\tau(X))}{\sigma_\tau} \right).$$

By setting $\mu_\tau \triangleq g_\tau$ and treating both $\mu_\tau$ and $\sigma_\tau$ as random variables that follow a prior distribution with state and action dependent hyperparameters, we can perform distributional reinforcement learning on a latent space. We can capture the randomness caused by the auto-encoding step with a probability measure $E_\phi(x, a, \tau)$, the parameters $\phi$ of which are trainable with a high-capacity function approximator. In our context, $\mu_\tau$ captures the mean quantile for these returns and $\sigma_\tau$ their standard deviation. The regression target $Y$ can be evaluated via Bellman backups.

The remaining two terms of the AIF ELBO perform a model update and policy entropy maximization. Whether to implement the latter is a design choice. We obtained better results with action-noise exploration than maximum entropy policy search (see Appendix B.1.4, Table 6). This is because the Inverse-Gamma prior on $\sigma_\tau$ in our uncertainty quantification pipeline acts as implicit epistemic uncertainty: the posterior variance over the scale parameter captures uncertainty in the quantile estimates, inducing Thompson-sampling-like stochasticity in the value function (Osband et al., 2013). Distributional losses additionally promote exploration via uncertainty-aware regularization (Sun et al., 2025). Adding

---

**Algorithm 3** Distributional Active Inference (DAIF)

**Input:** $P_A$
$x = \texttt{env.reset()}$
**while** True **do**
  $\pi \sim P_A(\cdot|x); \quad a := \pi(x); \quad x' := \texttt{env.step}(a)$
  $D := D \cup (x, a, x')$
  **repeat**
    $(x, a, x') \sim D, \ \tau \sim \mathcal{U}(0, 1), \ \pi \sim P_A(\cdot|x)$
    $\phi := \arg\max_{\phi'} \Big\{ \mathbb{E}_{\mu_\tau, \sigma^2_\tau \sim E_{\phi'}(x,a,\tau)} \big[ \log f_\tau \big($
        $R(x,a) + \gamma \mathbb{E}_{\mu'_\tau \sim E_{\phi'}(x', \pi(x'), \tau)} [\mu'_\tau] | \mu_\tau, \sigma^2_\tau) \big] \Big\}$
    $P_A := \arg\max_{P'_A} \Big\{ \mathbb{E}_{\pi \sim P'_A(\cdot|x)} \big[$
        $\mathbb{E}_{\mu_\tau \sim E_{\phi'}(x, \pi(x), \tau)} [\mu_\tau] \big] \big] + \mathbb{H}[P'_A] \Big\}$
  **until** end of training epoch
**end while**

---

SAC-style entropy regularization on top creates redundant exploration noise, as confirmed by our ablation (Table 6). As the model update is a lower bound to the same step of Algorithm 1, the same consequences of Lemma 3.1 apply.

With this we end up with Algorithm 3, which we refer to as *Distributional Active Inference (DAIF)*. This algorithm preserves the benefits of performing AIF whenever they are available in its native model-based version while eliminating the need to learn a transition kernel. A natural choice for $E_\phi$ would be the Normal-Inverse Gamma distribution with parameterized inputs. However, this would not permit an analytical calculation of the expectation of $\log p_\tau$ in the policy evaluation step. Having observed in our preliminary implementations that modeling the uncertainty around $\mu_\tau$ does not improve performance, we instead assumed a normal prior on the mean with fixed variance and modeled an Inverse Gamma prior for $\sigma_\tau$ (see Appendix B.2.4). The critic loss, after marginalizing over $\sigma$, takes the closed form:

$$\mathbb{E}_\sigma[\log f(G|\mu, \sigma, \tau)] = \log \tau(1-\tau) - \log \beta + \psi(\alpha)$$
$$- \frac{\alpha}{2\beta} (|G - \mu| + (2\tau - 1)(G - \mu)) \qquad (8)$$

where $G = R(x,a) + \gamma \mu'_{\tau'}$ is the Bellman target, $\alpha, \beta$ parameterize the Inverse-Gamma prior on $\sigma_\tau$, and $\psi$ is the digamma function. The full update procedure combines this objective with TD3-style design choices (Fujimoto et al., 2018): twin critics with min-clipping for the Belman target, action-noise exploration, delayed actor updates, and the weak hyperprior regularization of Akgül et al. (2025). See Appendix B.2.4 for the full derivation and the complete deep actor-critic update procedure (Algorithm 7).

## 5. Related Work

**Active inference and RL.** AIF casts control as inference,

selecting policies that minimize expected free energy (EFE), with epistemic terms inducing exploration (Friston et al., 2009; 2015). Most practical AIF agents are *model-based*: they learn an world model and use variational inference to maintain beliefs and/or evaluate EFE, often with planning (Ueltzhöffer, 2018; Çatal et al., 2019; Tschantz et al., 2020a; Fountas et al., 2020; Paul et al., 2021; Mazzaglia et al., 2021; Schneider et al., 2022). Complementary work analyzes links and limitations between AIF and RL/control-as-inference, including when EFE objectives recover RL-like optimality or reduce to standard intrinsic-motivation criteria (Millidge et al., 2020; Millidge, 2020; Sajid et al., 2021a; Da Costa et al., 2023). Closest to our work, Malekzadeh & Plataniotis (2024) derive Bellman-style EFE recursions for POMDPs and develop actor–critic updates in belief space, but still rely on learned belief representations and a world model. *DAIF* sidesteps world-model learning for EFE evaluation by learning targets directly from sampled transitions.

**Distributional RL.** Distributional RL models the return distribution rather than its expectation (Bellemare et al., 2017; Dabney et al., 2018b;a; Yang et al., 2019). It has been extended to actor–critic for continuous control (Barth-Maron et al., 2018; Ma et al., 2025) and applied to risk-sensitive control (Lim & Malik, 2022; Bernhard et al., 2019). Our push-forward RL framework generalizes distributional RL via push-forwards of trajectory measures, enabling integration with AIF. *Push-forward* also appears in Bai et al. (2025), but refers to transport-map parameterizations rather than our push-forwards within Bellman theory.

**Exploration in distributional RL.** Several works design exploration strategies within distributional RL (Tang & Agrawal, 2018; Mavrin et al., 2019; Zhou et al., 2021; Oh et al., 2022; Cho et al., 2023), while distributional losses have been shown to induce intrinsic exploration (Sun et al., 2025). In contrast, the exploration in DAIF arises naturally from variational free-energy minimization through the posterior uncertainty over the scale parameter $\sigma_\tau$, without requiring explicit exploration bonuses.

See Appendix C for an extended review.

# 6. Experiments

AIF postulates that the action-perception cycle of a biological agent is mediated by simulations on a latent space (Parr et al., 2022). Since the latent space applies state abstractions to maximize simulation efficiency, the advantages of AIF should appear where state abstractions facilitate learning.

**Tabular experiments.** We evaluate on a grid world, *Latent RiverSwim*, whose transition probabilities share RiverSwim's dynamics (Osband et al., 2013) on a one-

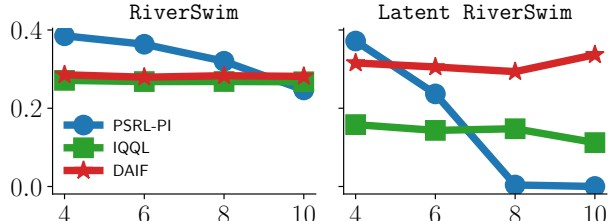

*Figure 2.* **Horizontal axis:** Distance from the initial state to the most desired state. **Vertical axis:** Frequency of the visitation of the most desired state. DAIF matches the plain distributional RL performance when transition dynamics cannot be represented more efficiently in a latent space (left panel). DAIF outperforms both distributional and model-based counterparts when a latent manifold drives the dynamics at a degree increasing with the difficulty of the problem (right panel). For the learning curves of individual configurations, see Figure 5 in the appendix.

dimensional latent manifold unknown to the agent. The agent observes a 2D state but the reward-relevant dynamics live on a 1D manifold defined by a linear encoder $e_\alpha(i,j) = \alpha i + (1 - \alpha)j$. As the horizon grows, the observation space scales quadratically while the latent space grows linearly, making state abstraction beneficial. See Appendix B.1 and Figure 4 for details. We compare DAIF against two tabular baselines: PSRL-PI (Osband et al., 2013), a policy-iteration variant of posterior sampling representative of model-based RL, and IQQL, a tabular implicit quantile network representative of distributional RL (Dabney et al., 2018a). Figure 2 (right) demonstrates that both model-based RL and distributional RL fail to solve the task as the horizon grows, whereas DAIF maintains performance. As shown in the left panel, the models behave similarly in long horizons when state abstractions are not advantageous.[1]

**Continuous control experiments.** We adapt DAIF to continuous control with deep actor-critics (Appendix B.2.4; the full update procedure is given in Algorithm 7). We evaluate on three benchmark suites: (i) *EvoGym* (Bhatia et al., 2021), featuring soft robots across locomotion and manipulation tasks (7 environments); (ii) *DeepMind Control Suite (DMC)* (Tassa et al., 2018), providing continuous control on MuJoCo locomotors with varying morphology and dimensionality (7 environments); and (iii) *DMC Vision*, where the same DMC tasks are solved from raw pixel observations (5 environments). Our deep-RL baselines are DRND (Yang et al., 2024), an exploration-driven model-free method; DSAC (Ma et al., 2025), the state-of-the-art distributional actor-critic; DTD3, a distributional extension of TD3 (Fujimoto et al., 2018) that we introduce as a direct ablation removing only the AIF component; and, for DMC vision, DrQ-v2 (Yarats et al., 2022). As summarized in

---

[1]We choose the policy iteration variant of PSRL due to its competitive empirical performance (Tiapkin et al., 2022, Figure 1), although its regret profile is not yet fully characterized.

*Table 1.* Ranking comparison on 19 continuous control environments from three benchmark suites. The algorithms are ranked separately for each repetition of each environment. The sign ± indicates the standard deviation. Area Under the Learning Curve (AULC) measures sample efficiency and Final Return measures how well a control task has been solved. The method with the smallest rank value is marked in bold. As the architecture and hyperparameters of the original DrQ-v2 (Yarats et al., 2022) is tuned specifically for visual control, we do not evaluate this model in the other two benchmark suites. **# Envs:** Number of environments. **# Reps:** Number of experiment replications.

| Suite | # Envs | # Reps | Area Under the Learning Curve (AULC) | | | | | Final Return | | | | |
|---|---|---|---|---|---|---|---|---|---|---|---|---|
| | | | DRND | DRQv2 | DSAC | DTD3 | DAIF | DRND | DRQv2 | DSAC | DTD3 | DAIF |
| EvoGym | 7 | 10 | 3.2 ±1.1 | — | 2.9 ±0.8 | 2.0 ±0.8 | **1.5** ±0.7 | 3.7 ±0.7 | — | 2.8 ±0.8 | 2.0 ±0.8 | **1.6** ±0.8 |
| DMC | 7 | 10 | 3.2 ±1.1 | — | 2.8 ±0.9 | 2.4 ±1.0 | **1.6** ±0.7 | 3.1 ±1.2 | — | 2.8 ±0.8 | 2.6 ±0.9 | **1.5** ±0.8 |
| DMC Vision | 5 | 5 | 4.2 ±1.3 | 3.5 ±1.1 | 2.5 ±1.1 | 2.9 ±1.2 | **1.9** ±1.2 | 4.4 ±1.0 | 3.0 ±1.2 | 2.8 ±1.1 | 3.8 ±1.2 | **2.0** ±1.4 |

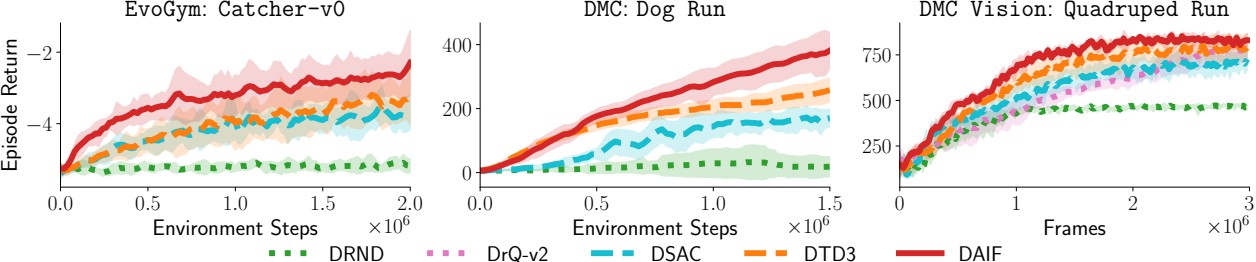

*Figure 3.* Evaluation curves for three representative environments, one per suite, where DAIF clearly improves the state of the art. These are relatively harder problems of the related suite due to either complex dynamics or large state or action dimensionality, where the abstraction of the return distribution is beneficial. DAIF performs comparably to the state of the art when it does not improve. For the learning curves of the remaining environments, see Appendix B.2.

Table 1, DAIF provides consistent performance gains across all three suites. Controlling DMC locomotors is difficult due to compounding function-approximation errors that cause the deadly triad (Ciosek et al., 2019). DMC vision presents the challenge of controlling these platforms from the high-dimensional state space comprising raw pixels. EvoGym introduces control problems for soft robots.

Figure 3 reports learning curves for one representative environment per suite. *Catcher-v0* is classified as a *hard* task in EvoGym (Bhatia et al., 2021) and has the steepest learning curve within the interaction budget. *Dog Run* has the highest state dimensionality among DMC robots, and running demands sustained high-speed control. *Quadruped Run* has the highest action dimensionality among the vision-based environments. DAIF outperforms state-of-the-art baselines by a clear margin throughout the learning process. DAIF's performance boost corroborates our view that AIF is useful for controlling challenging environments with limited computational resources. The performance boost of DAIF requires about 12% more wall-clock computation time than distributional actor-critic methods, which is less than the overhead of DSAC (26%) and DRND (37%) relative to DTD3. See Appendix B.2 for details of these experiments.

## 7. Takeaways and Open Questions

We provided a new theoretical framework that enables casting AIF as a simple extension of distributional RL. This implies that the improvements in the adaptation capabilities

of an AIF agent should be observed in the distributional setting. We reported a comprehensive set of experiment results that support this claim. The deep RL implementation of DAIF improved the state of the art in multiple soft robotics benchmarks and challenging visual control tasks.

DAIF inherits the convergence guarantees of distributional RL with an improved contraction modulus. A rigorous characterization of its finite-sample characteristics is an open question, as it is for the distributional RL field. Our theoretical framework can also lay a foundation for a formal analysis of AIF's computational properties. A methodology for this could be extending the sample complexity analysis of linear quadratic control (Krauth et al., 2019) to well-behaved non-linearities.

**Limitations as a description of active inference.** While DAIF integrates AIF into distributional RL, it does not implement an explicit EFE objective or separate epistemic and instrumental value terms. The epistemic component appears implicitly through the posterior uncertainty of the Inverse-Gamma prior on $\sigma_\tau$, rather than through an explicit information-gain bonus as in canonical AIF. Furthermore, the size of the latent bottleneck—which controls the $L_E \cdot L_D$ trade-off in Theorem 3.5—is set via the architecture (hidden dimensionality) rather than optimized automatically. An interesting direction for future work is to learn the bottleneck size adaptively, potentially via architecture search or information-theoretic regularization.

## Impact Statement

This work makes both theoretical and algorithmic contributions to the general field of machine learning. The theoretical contributions are two-fold. The first is a formulation of the key concepts of distributional reinforcement learning using only stochastic process constructions and their transformations. The second is a derivation of the active inference machinery using variational and causal inference concepts in a rigorous and consistent manner. This derivation highlights a significant simplification in the resulting objective function. The algorithmic contribution is a distributional counterpart of active inference, which has thus far been practiced in computationally demanding model-based settings. We do not foresee negative societal impacts beyond those generally associated with increased automation.

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

# Appendix

*Table 2.* Notation used throughout the paper.

| | |
|---|---|
| $\triangleq$ | Definition of a variable. |
| $:=$ | Assignment of an already defined variable to a new value. |
| $(g \circ h)(x) \triangleq g(h(x))$ | Composition of functions $g$ and $h$ for some input $x$. |
| $g^{-1}(B) \triangleq \{x \in \mathcal{X} : g(x) \in B\}$ | Pre-image of set $B$ under function $g$. |
| $\|g(x)\|_\infty \triangleq \sup_{x \in \mathcal{X}} |g(x)|$ | Supremum norm. |
| $\mathbf{1}(\rho)$ | Indicator function; returns 1 if predicate $\rho$ is true, 0 otherwise. |
| $\mathrm{sign}(x)$ | Sign function that returns $+1$ if $x \geq 0$ and $-1$ otherwise. |
| $\mathbb{N}$ | The set of natural numbers $\{0, 1, 2, \ldots\}$. |
| $\mathbb{N}_+$ | The set of strictly positive natural numbers $\{1, 2, 3, \ldots\}$. |
| $\mathbb{R}$ | The set of real numbers. |
| $\mathbb{R}_+$ | The set of positive real numbers. |
| $\sigma(\mathcal{X})$ | The $\sigma$-algebra generated by $\mathcal{X}$. |
| $\mathcal{B}(\mathbb{R})$ | Borel $\sigma$-algebra on $\mathbb{R}$. |
| $(\mathcal{X}, \sigma(\mathcal{X}))$ | Measurable space with $\sigma$-algebra $\sigma(\mathcal{X})$. |
| $(\mathcal{X}, \sigma(\mathcal{X}), P)$ | Probability space constructed by measuring $(\mathcal{X}, \sigma(\mathcal{X}))$ with $P$. |
| $\mathcal{P}_\mathcal{X}$ | The set of all probability measures on $(\mathcal{X}, \sigma(\mathcal{X}))$. |
| $\delta_x(A) \triangleq \mathbf{1}(x \in A)$ | Dirac measure evaluated at $x$ for some measurable set $A$. |
| $F_P(x) \triangleq P(X \leq x)$ | Cumulative Distribution Function (CDF) of measure $P$. |
| $f_P(x) \triangleq dF_P(x)/dx$ | Probability Density Function (PDF) of measure $P$. |
| $\mathbb{E}_{x \sim P}[g(x)] \triangleq \int_\mathcal{X} g(x) P(dx)$ | Expected value of measurable function $g$ with respect to probability measure $P$. |
| $\mathbb{V}_{x \sim P}[g(x)] \triangleq \int_\mathcal{X} g^2(x) P(dx) - \left(\int_\mathcal{X} g(x) P(dx)\right)^2$ | Variance of measurable function $g$ with respect to probability measure $P$. |
| $P_{X|Y} P_{Y|z} \triangleq \int P_{X|Y}(\cdot|Y) P_{Y|Z}(dY|z) = P_{X|z}$ | Marginalization of an intermediate random variable $Y$ conditioned on a point observation $z$. |
| $P_X P_Y|z \triangleq \int P_X(\cdot|Y) P_{Y|Z}(dY|z) = P_{X|z}$ | Shorthand where the conditioning on $Y$. |
| $P_X P_Y \triangleq \int P_X(\cdot|Y) P_Y(dY)$ | Shorthand for unconditional marginalization. |
| $\mathcal{U}(a, b)$ | Continuous uniform distribution defined on range $a < b \in \mathbb{R}$. |
| $\mathcal{U}(A)$ | Discrete distribution where each element of set $A$ gets equal probability, i.e., $P(X = a) = 1/|A|, \forall a \in A$. |
| $\mathcal{IG}(\sigma|\alpha, \beta) = \frac{\beta^\alpha}{\Gamma(\alpha)} \sigma^{-\alpha-1} \exp\left(-\beta/\sigma\right)$ | Inverse Gamma distribution with shape $\alpha$ and scale $\beta$. |
| $\mathrm{Dirichlet}(\boldsymbol{\alpha})$ | Dirichlet distribution with concentration parameters $\boldsymbol{\alpha} = (\alpha_1, \ldots, \alpha_K), \alpha_i > 0$. |
| $\Gamma(x)$ | Gamma function, the continuous extension of the factorial with $\Gamma(n) = (n-1)!$ for positive integers. |
| $\psi(x)$ | Digamma function, the logarithmic derivative of the gamma function: $\psi(x) = \frac{d}{dx} \log \Gamma(x)$. |
| $\mathbb{H}[P] \triangleq -\int_\mathcal{X} \log p(x) P(dx)$ | (Differential) entropy. |
| $KL(P\|\bar{P}) \triangleq \int_\mathcal{X} \log\left(\frac{dP}{d\bar{P}}\right) P(dx)$ | Kullback-Leibler divergence between probability measures $P$ and $\bar{P}$. |
| $\Gamma(P, \bar{P})$ | The set of couplings between probability measures $P$ and $\bar{P}$. |
| $\mathcal{W}_p(P, \bar{P}) \triangleq \left(\inf_{\nu \in \Gamma(P, \bar{P})} \mathbb{E}_{(x, x') \sim \nu}[|x - x'|^p]\right)^{1/p}$ | $p$-Wasserstein distance for absolute norm. |
| $\bar{\mathcal{W}}_p(P_{x,a}, \bar{P}_{x,a}) \triangleq \sup_{x,a} \mathcal{W}_p(P_{x,a}, \bar{P}_{x,a})$ | Maximal form of $p$-Wasserstein distance. |
| $P_{W|\mathrm{do}(X \sim P(\cdot))}$ | do-operator that replaces the distribution of variable $X$ in a structural causal model $W$ by $P$. |

# A. Proofs and derivations

### A.1. Proof of Lemma 3.1

For a fixed $(x, a)$ the Wasserstein distance is regular, homogeneous, and $p$-convex (see, e.g., Bellemare et al., 2023, Definition 4.22-4.24 for details), i.e.,

$$
\begin{aligned}
\mathcal{W}_p^p \left( \mathbf{T}_{P_*}^\pi \mathbf{F}_\# \mathbb{P}_{x,a}^{P_\pi}, \mathbf{T}_{P_*}^\pi \mathbf{F}_\# \mathbb{P}_{x,a}^{\bar{P}_\pi} \right) &\leq \gamma^p \mathcal{W}_p \left( \mathbb{E}_{x' \sim P_\pi^*(x'|x,\pi(x))} \left[ \mathbf{F}_\# \mathbb{P}_{x',\pi(x)}^{P_\pi} \right], \mathbb{E}_{x' \sim P_\pi^*(x'|x,\pi(x))} \left[ \mathbf{F}_\# \mathbb{P}_{x',\pi(x)}^{\bar{P}_\pi} \right] \right) \\
&\leq \gamma^p \mathbb{E}_{x' \sim P_\pi^*(x'|x,\pi(x))} \left[ \mathcal{W}_p \left( \mathbf{F}_\# \mathbb{P}_{x',\pi(x)}^{P_\pi}, \mathbf{F}_\# \mathbb{P}_{x',\pi(x)}^{\bar{P}_\pi} \right) \right] \\
&\leq \gamma^p \sup_{x',a'} \mathcal{W}_p^p \left( \mathbf{F}_\# \mathbb{P}_{x',\pi(x)}^{P_\pi}, \mathbf{F}_\# \mathbb{P}_{x',\pi(x)}^{\bar{P}_\pi} \right) = \gamma^p \bar{\mathcal{W}}_p^p \left( \mathbf{F}_\# \mathbb{P}^{P_\pi}, \mathbf{F}_\# \mathbb{P}^{\bar{P}_\pi} \right),
\end{aligned}
$$

where the first inequality follows, via regularity and homogeneity, the second via $p$-convexity, the third via the definition of a supremum and finally the equality via the definition of $\bar{\mathcal{W}}_p$. As this holds for every pair $(x, a)$ taking the $p$-th root and subsequently the supremum on the left hand side gives the desired inequality $\square$

### A.2. Proof of Lemma 3.4

By definition, $(KF)(x, \cdot) = K(F(x), \cdot)$. Using the $L$-Lipschitz property of the kernel $K$ on $\mathcal{S}$, we have:

$$
\mathcal{W}_p(K(F(x), \cdot), K(F(x'), \cdot)) \leq L \cdot |F(x) - F(x')|.
$$

Applying the $M$-Lipschitz property of $F$, it follows that $|F(x) - F(x')| \leq M \cdot |x - x'|$. Substituting this into the inequality above gives:

$$
\mathcal{W}_p((KF)(x, \cdot), (KF)(x', \cdot)) \leq L \cdot M \cdot |x - x'| \quad \square
$$

### A.3. Proof of Theorem 3.5

We apply a cascade of the following two simple results. The first is that passing a Markov kernel through an $L$-Lipschitz operator is a contraction with respect to $p$-Wasserstein distance.

**Lemma A.1.** *For every $P, P' \in \mathcal{P}_\mathcal{X}$ and $L$-Lipschitz Markov kernel $K$, the following holds:*

$$
\mathcal{W}_p(KP, KP') \leq L \cdot \mathcal{W}_p(P, P').
$$

*Proof.* Let $\pi$ be an optimal coupling for $(P, P')$ with respect to $d_p$. For every pair $(s, s') \sim \pi$ let $\gamma_{s,s'}$ be the optimal coupling of $(K(s, \cdot), K(s', \cdot))$. We then have

$$
\begin{aligned}
\mathcal{W}_p^p(KP, KP') &\leq \mathbb{E}_{(s,s') \sim \pi} \left[ \mathcal{W}_p^p(K(s, \cdot), K(s', \cdot)) \right] \\
&\leq \mathbb{E}_{(s,s') \sim \pi} \left[ L \cdot |s - s'|^p \right] \\
&= L \cdot \mathbb{E}_{(s,s') \sim \pi} \left[ |s - s'|^p \right] \\
&= L \cdot \mathcal{W}_p^p(P, P').
\end{aligned}
$$

The first inequality follows via the definition of the Wasserstein distance as the infimum over all couplings, where the right hand side is the expected distance for the chosen coupling strategy (pairing via $\pi$, then $\gamma_{s,s'}$). The second inequality follows via the Lipschitz assumption and the final via the choice of $\pi$ as the optimal coupling of $(P, P')$. Taking the $p$-th root gives the desired inequality. $\square$

The second is that passing the Markov kernels of two Markov processes through an $L$-Lipschitz operator contracts their $p$-Wasserstein distance by a factor $L$.

**Lemma A.2.** *Let $\mathbb{P}_{x,a}^P$ and $\mathbb{P}_{x,a}^{P'}$ be the laws of two Markov processes generated by kernels $P$ and $P'$ respectively. Let $K$ be an $L$-Lipschitz probabilistic operator. Define the transformed processes $\mathbb{P}_{x,a}^{KP}$ and $\mathbb{P}_{x,a}^{KP'}$ as the laws generated by the composed kernels $KP$ and $KP'$. Then we have*

$$
\mathcal{W}_p(\mathbb{P}_{x,a}^{KP}, \mathbb{P}_{x,a}^{KP'}) \leq L \cdot \mathcal{W}_p(\mathbb{P}_{x,a}^P, \mathbb{P}_{x,a}^{P'}).
$$

*Proof.* Let $\nu$ be an optimal coupling of the path measures $(\mathbb{P}_{x,a}^P, \mathbb{P}_{x,a}^{P'})$ such that the total cost satisfies $\mathbb{E}_\nu[\sum_{t=0}^\infty |X_t - X_t'|^p] = \mathcal{W}_p^p(\mathbb{P}_{x,a}^P, \mathbb{P}_{x,a}^{P'})$. By the definition of the Wasserstein distance on the product space and the $L$-Lipschitz property of $K$, we have for each time step $t$:

$$\mathcal{W}_p^p(K(X_t, \cdot), K(X_t', \cdot)) \le L^p \cdot |X_t - X_t'|^p.$$

Summing over the horizon and taking the limit $T \to \infty$, we bound the distance between the transformed processes by integrating the point-wise kernel distances over the path coupling $\nu$:

$$\mathcal{W}_p^p(\mathbb{P}_{x,a}^{KP}, \mathbb{P}_{x,a}^{KP'}) \le \lim_{T\to\infty} \mathbb{E}_\nu\left[\sum_{t=0}^T \mathcal{W}_p^p(K(X_t, \cdot), K(X_t', \cdot))\right]$$

$$\le \lim_{T\to\infty} \mathbb{E}_\nu\left[\sum_{t=0}^T L^p \cdot |X_t - X_t'|^p\right]$$

$$= L^p \cdot \mathbb{E}_\nu\left[\sum_{t=0}^\infty |X_t - X_t'|^p\right]$$

$$= L^p \cdot \mathcal{W}_p^p(\mathbb{P}_{x,a}^P, \mathbb{P}_{x,a}^{P'}).$$

Taking the $p$-th root yields the desired result. $\qquad\square$

*Proof of Theorem 3.5.* First, we observe that the transition kernel $(P_D S) : \mathcal{X} \to \mathcal{P}_\mathcal{X}$ is a composition of an $L_E$-Lipschitz map $S$ and an $L_D$-Lipschitz kernel $P_D$. Consequently, $(P_D S)$ is an $(L_D L_E)$-Lipschitz Markov kernel in the $p$-Wasserstein sense.

Applying Lemma A.2, the distance between the path-space measures generated by the transformed kernels is bounded by:

$$\mathcal{W}_p\left(\mathbb{P}^{(P_D S)P_\pi}, \mathbb{P}^{(P_D S)\bar{P}_\pi}\right) \le (L_D L_E) \cdot \mathcal{W}_p\left(\mathbb{P}^{P_\pi}, \mathbb{P}^{\bar{P}_\pi}\right).$$

By the property of the process-level push-forward functional $\mathbf{F}_\#$, this relationship is preserved. Finally, applying the $\gamma$-contraction property of the operator $\mathbf{T}_{P_*}^\pi$, we obtain:

$$\bar{\mathcal{W}}_p\left(\mathbf{T}_{P_*}^\pi \mathbf{F}_\# \mathbb{P}^{(P_D S)P_\pi}, \mathbf{T}_{P_*}^\pi \mathbf{F}_\# \mathbb{P}^{(P_D S)\bar{P}_\pi}\right)$$

$$\le \gamma \cdot \bar{\mathcal{W}}_p\left(\mathbf{F}_\# \mathbb{P}^{(P_D S)P_\pi}, \mathbf{F}_\# \mathbb{P}^{(P_D S)\bar{P}_\pi}\right)$$

$$\le \gamma \cdot L_E \cdot L_D \cdot \bar{\mathcal{W}}_p\left(\mathbf{F}_\# \mathbb{P}^{P_\pi}, \mathbf{F}_\# \mathbb{P}^{\bar{P}_\pi}\right).$$

The result follows. $\qquad\square$

## A.4. Factorization of the intervened world model

Define

$$\widetilde{P}(X, Y, S) \triangleq P_D(X|Y, S)P_Q(Y, S). \tag{9}$$

Then by product rule

$$\widetilde{P}(X, Y, S)_{\mathrm{do}(X \sim P_R)} = \widetilde{P}(X|Y, S)_{\mathrm{do}(X \sim P_R)}\widetilde{P}(Y, S)_{\mathrm{do}(X \sim P_R)}$$

$$= \widetilde{P}(X|Y, S)_{\mathrm{do}(X \sim P_R)}\widetilde{P}(Y, S)$$

$$= P_R(X)\widetilde{P}(Y, S)$$

By definition we have

$$\widetilde{P}(X, Y, S)_{\mathrm{do}(X \sim P_R)} = (P_D)_{\mathrm{do}(X \sim P_R)}(X|Y, S)(P_Q)_{\mathrm{do}(X \sim P_R)}(Y, S)$$

$$= P_R(X)P_Q(Y,S).$$

which yields $\widetilde{P}(Y,S) = P_Q(Y,S)$ hence

$$\widetilde{P}(X,Y,S)_{\mathrm{do}(X\sim P_R)} = P_R(X)P_Q(Y,S).$$

Therefore

$$\widetilde{P}(Y,S|X)_{\mathrm{do}(X\sim P_R)} = P_Q(Y,S).$$

## A.5. Derivation of the active inference objective

Consider that

$$
\begin{aligned}
P_{\widetilde{W}|\mathrm{do}(Y\sim\delta_y,S\sim P_Q)}(X,Y,S) &= P_{\widetilde{W}|\mathrm{do}(Y\sim\delta_y)}(X,Y,S) \\
&= (P_Q(Y,S)P_R(X))_{\mathrm{do}(Y\sim\delta_y)} \\
&= (P_Q(Y,S))_{\mathrm{do}(Y\sim\delta_y)}P_R(X) \\
&= P_Q(y,S)P_R(X).
\end{aligned}
$$

The first equality follows from the fact that $P_{\widetilde{W}}$ has already been constructed with the intervention $\mathrm{do}(S \sim P_Q)$ and the second is due to the result shown in Appendix A.4. Now the related AIF posterior can be constructed as follows

$$
\begin{aligned}
\log f_{P_0}(Y := y) + \mathrm{const} &= \mathbb{E}_{x',s\sim P_{W|\mathrm{do}(Y\sim\delta_y,S\sim P_Q)}}\Big[L(x', P_{\widetilde{W}|\mathrm{do}(Y\sim\delta_y)}, P_Q)\Big] \\
&= \mathbb{E}_{x',s\sim P_{W|\mathrm{do}(Y\sim\delta_y,S\sim P_Q)}}\Big[\mathbb{E}_{y',s'\sim P_Q}\Big[\log\Big(P_Q(y,s')P_R(x')/P_Q(y',s')\Big)\Big]\Big] \\
&= \mathbb{E}_{x',s\sim P_{W|\mathrm{do}(Y\sim\delta_y,S\sim P_Q)}}\Big[\log P_R(x') + \mathbb{E}_{y',s'\sim P_Q}\Big[\log\Big(P_Q(y,s')/P_Q(y',s')\Big)\Big]\Big] \\
&= \mathbb{E}_{x'\sim P_D P_{Q|y}}\Big[\log P_R(x')\Big] + \mathbb{E}_{x'\sim P_D P_{Q|y}}\Big[\mathbb{E}_{y',s'\sim P_Q}[\log P_Q(y,s')] - \mathbb{E}_{y',s'\sim P_Q}[\log P_Q(y',s')]\Big]
\end{aligned}
$$

where $\mathrm{const}$ represents the normalization constant that does not depend on $Y$. Now let us take the expectation with respect to the posterior on $Y$:

$$
\begin{aligned}
&\mathbb{E}_{y\sim P_Q}[\log f_{P_0}(Y := y)] + \mathrm{const} \\
&= \mathbb{E}_{y\sim P_Q}\Bigg[\mathbb{E}_{x'\sim P_D P_{Q|y}}\Big[\log P_R(x') + \mathbb{E}_{x'\sim P_D P_{Q|y}}\Big[\mathbb{E}_{y',s'\sim P_Q}[\log P_Q(y,s')] - \mathbb{E}_{y',s'\sim P_Q}[\log P_Q(s',y')]\Big]\Big]\Bigg] \\
&= \mathbb{E}_{x'\sim P_D P_Q}[\log P_R(x')] + \mathbb{E}_{y\sim P_Q}\Bigg[\mathbb{E}_{x'\sim P_D P_{Q|y}}\Big[\mathbb{E}_{y',s'\sim P_Q}[\log P_Q(y,s')] - \mathbb{E}_{y',s'\sim P_Q}[\log P_Q(y',s')]\Big]\Bigg] \\
&= \mathbb{E}_{x'\sim P_D P_Q}[\log P_R(x')] + \mathbb{E}_{y\sim P_Q}\Bigg[\mathbb{E}_{x'\sim P_D P_{Q|y}}\Big[\mathbb{E}_{y',s'\sim P_Q}[\log P_Q(y,s')]\Big]\Bigg] - \mathbb{E}_{y\sim P_Q}\Bigg[\mathbb{E}_{y',s'\sim P_Q}[\log P_Q(y',s')]\Bigg] \\
&= \mathbb{E}_{x'\sim P_D P_Q}[\log P_R(x')] + \mathbb{E}_{y\sim P_Q}\Bigg[\mathbb{E}_{x'\sim P_D P_{Q|y}}\Big[\mathbb{E}_{y',s'\sim P_Q}[\log P_Q(y,s')]\Big]\Bigg] - \mathbb{E}_{y',s'\sim P_Q}[\log P_Q(y',s')] \\
&= \mathbb{E}_{x'\sim P_D P_Q}[\log P_R(x')] + \mathbb{E}_{x',y\sim P_D P_Q}\Big[\mathbb{E}_{y',s'\sim P_Q}[\log P_Q(y,s')]\Big] - \mathbb{E}_{y',s'\sim P_Q}[\log P_Q(y',s')] \\
&= \mathbb{E}_{x'\sim P_D P_Q}[\log P_R(x')] + \mathbb{E}_{x',y'\sim P_D P_Q}\Big[\mathbb{E}_{y,s'\sim P_Q}[\log P_Q(y,s')]\Big] - \mathbb{E}_{y',s'\sim P_Q}[\log P_Q(y',s')] \\
&= \mathbb{E}_{x'\sim P_D P_Q}[\log P_R(x')] + \mathbb{E}_{y,s'\sim P_Q}[\log P_Q(y,s')] - \mathbb{E}_{y',s'\sim P_Q}[\log P_Q(y',s')] \\
&= \mathbb{E}_{x'\sim P_D P_Q}[\log P_R(x')].
\end{aligned}
$$

Plugging this result into the ELBO, we get

$$L(x, P_W, P_Q) = \mathbb{E}_{y,s\sim P_Q}\Big[\log\Big(P_D(x|y,s)P_0(y,s)/P_Q(y,s)\Big)\Big]$$

$$\begin{aligned}
&= \mathbb{E}_{y,s\sim P_Q}[\log P_D(x|y,s)] + \mathbb{E}_{y,s\sim P_Q}[\log P_0(y,s)] + \mathbb{H}[P_Q] \\
&= \mathbb{E}_{y,s\sim P_Q}[\log P_D(x|y,s)] + \mathbb{E}_{y,s\sim P_Q}[\log P_0(s|y)] + \mathbb{E}_{y\sim P_Q}[\log P_0(y)] + \mathbb{H}[P_Q] \\
&= \mathbb{E}_{y,s\sim P_Q}[\log P_D(x|y,s)] + \mathbb{E}_{y,s\sim P_Q}[\log P_0(s|y)] + \mathbb{E}_{x'\sim P_D P_Q}[\log P_R(x')] + \mathbb{H}[P_Q].
\end{aligned}$$

## A.6. Derivation of the upper bound to the push-forward RL objective

$$\begin{aligned}
J(S) &= \mathbb{E}_{x\sim\mathcal{U}(\mathcal{X})}[d_2^2((\mathbf{G}_{x_0,a_0}^\pi)_\#\mathbb{P}_{x_0,a_0}^{(P_D S)P_\pi}, \mathbf{T}_{P_*}^\pi(\mathbf{G}_{x',\pi(x')}^\pi)_\#\mathbb{P}_{x_0,a_0}^{(P_D S)P_\pi})] \\
&\leq \mathbb{E}_{x\sim\mathcal{U}(\mathcal{X})}[\mathbb{E}_{\mathbf{s}\sim\mathbf{S}_\#\mathbb{P}_{x_0,a_0}^{P_\pi}}[d_2^2((\mathbf{G}_{x_0,a_0}^\pi)_\#\mathbb{P}_{x_0,a_0}^{P_D|\mathbf{s}}(B), \mathbf{T}_{P_*}^\pi\mathbf{G}_{x',\pi(x')}^\pi)_\#\mathbb{P}_{x_0,a_0}^{P_D|\mathbf{s}}(B))]] \\
&= \mathbb{E}_{x\sim\mathcal{U}(\mathcal{X})}\left[\mathbb{E}_{\mathbf{s}\sim\mathbf{S}_\#\mathbb{P}_{x_0,a_0}^{P_\pi}}\left[\int_0^1\left(F_{(\mathbf{G}_{x_0,a_0}^\pi)_\#\mathbb{P}_{x_0,a_0}^{P_D|\mathbf{s}}}^{-1}(\tau) - F_{\mathbf{T}_{P_*}^\pi\mathbf{G}_{x',\pi(x')}^\pi)_\#\mathbb{P}_{x_0,a_0}^{P_D|\mathbf{s}}}^{-1}(\tau)\right)^2 d\tau\right]\right] \\
&= \mathbb{E}_{x\sim\mathcal{U}(\mathcal{X})}\left[\mathbb{E}_{\mathbf{s}\sim\mathbf{S}_\#\mathbb{P}_{x_0,a_0}^{P_\pi}}\left[\mathbb{E}_{\tau\in\mathcal{U}(0,1)}\left[\left(\left(F_{(\mathbf{G}_{x_0,a_0}^\pi)_\#\mathbb{P}_{x_0,a_0}P_D|\mathbf{s}}^{-1}(\tau) - F_{\mathbf{T}_{P_*}^\pi\mathbf{G}_{x',\pi(x')}^\pi)_\#\mathbb{P}^{P_D|\mathbf{s}}}^{-1}(\tau)\right)\right)^2\right]\right]\right]
\end{aligned}$$

where the inequality follows from the composite nature of $(\mathbf{G}_{x_0,a_0}^\pi)_\#\mathbb{P}_{x_0,a_0}^{(P_D S)P_\pi}$ established in Eq. 5. The result follows straightforwardly from the role of couplings in the $p$-Wasserstein distance formulation and it is commonly used by prior work in other contexts such as Wasserstein Auto Encoders (Tolstikhin et al., 2018).

## A.7. Construction of Markov process measures

Define a cylinder set of length $n$ for measurable sets $B_1, B_2, \ldots, B_n \in \sigma(\mathcal{X})$ as

$$C = \{(x_1, x_2, \ldots) \in \mathcal{X}^{\mathbb{N}_+} : x_1 \in B_1, x_2 \in B_2, \ldots, x_n \in B_n\}.$$

According to the Ionescu-Tulcea theorem (Ionescu-Tulcea, 1949), there exists a unique probability measure $\mathbb{P}_{x_0,a_0}^{P^\pi}$ defined on the product $\sigma$-algebra $\sigma(\mathcal{X})^{\otimes\mathbb{N}_+}$ such that for every $n \in \mathbb{N}_+$ and every cylinder set $C$ of the form $B_1 \times B_2 \times \cdots \times B_n \times \mathcal{X} \times \cdots$, with $B_t \in \sigma(\mathcal{X})$ for $t \geq 1$, the measure is consistent with

$$\mathbb{P}_{x_0,a_0}^{P^\pi}(C) = \int_{B_1} P(\mathrm{d}x_1|x_0,a_0) \times \int_{B_2} P^\pi(\mathrm{d}x_2|x_1) \times \int_{B_3} P^\pi(\mathrm{d}x_3|x_2) \times \cdots \times \int_{B_n} P^\pi(\mathrm{d}x_n|x_{n-1}).$$

This measure defines the canonical process $(X_n)_{n\geq 1}$ on the path space $\mathcal{X}^{\mathbb{N}_+}$, where $X_n(\omega) \triangleq \omega_n$ for all $n \geq 1$ and $\omega_n$ denotes the $n$'th coordinate of the trajectory. The resulting triple $(\mathcal{X}^{\mathbb{N}_+}, \sigma(\mathcal{X})^{\otimes\mathbb{N}_+}, \mathbb{P}_{x_0,a_0}^{P^\pi})$ constitutes a valid probability space for the infinite-horizon controllable Markov process that starts from $P(\cdot|x_0,a_0)$ and proceeds with the policy-induced kernel $P^\pi$.

# B. Experiment Details

## B.1. Tabular environments

### B.1.1. RIVERSWIM

The RiverSwim (Osband et al., 2013) environment is defined over a one-dimensional state space $\mathcal{X} = \{1, \ldots, n\}$ with binary actions $\mathcal{A} = \{-1, +1\}$. The transition dynamics are parameterized by probabilities $p_{\text{forward}}$ and $p_{\text{backward}}$ and are given by

$$\begin{aligned}
P(X' = k+1|X = k, A = +1) &= p_{\text{forward}}\mathbf{1}(2 \leq k \leq n-1) + \left(1 - (p_{\text{forward}} + p_{\text{backward}})\right)\mathbf{1}(k=1), \\
P(X' = k|X = k, A = +1) &= \left(1 - (p_{\text{forward}} + p_{\text{backward}})\right)\mathbf{1}(k \geq 2) + (p_{\text{forward}} + p_{\text{backward}})\mathbf{1}(k=1), \\
P(X' = k-1|X = k, A = +1) &= p_{\text{backward}}\mathbf{1}(2 \leq k \leq n-1) + (p_{\text{forward}} + p_{\text{backward}})\mathbf{1}(k=n), \\
P(X' = k+1|X = k, A = -1) &= 0, \\
P(X' = k|X = k, A = -1) &= \mathbf{1}(k=1), \\
P(X' = k-1|X = k, A = -1) &= \mathbf{1}(k-1 \geq 1).
\end{aligned}$$

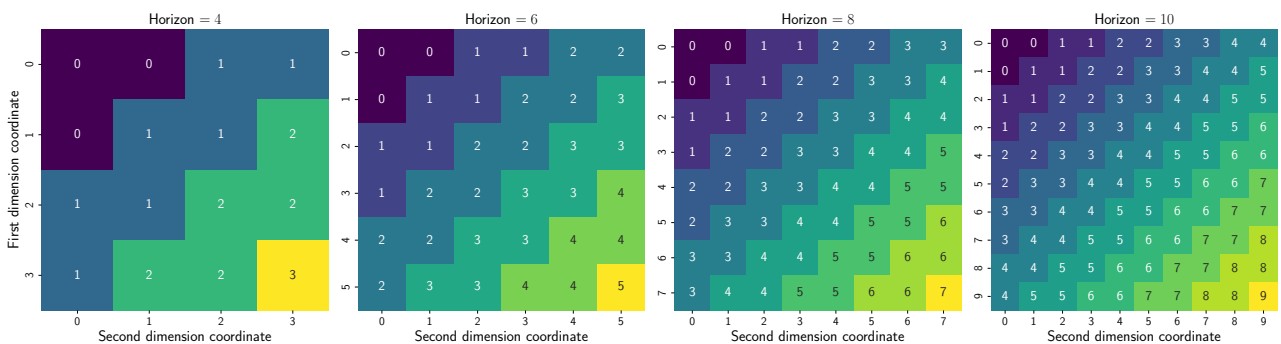

*Figure 4.* Observation-to-latent state mappings in the Latent RiverSwim environment across multiple horizons. The $(i, j)$ coordinates represent indices for each observation in $\mathcal{X}$, while cell values and colours indicate the corresponding latent state index from $\mathcal{S}$. For instance, at Horizon = 4, the cell $(i = 3, j = 1)$ has value 2, signifying that observation $(3, 1)$ is mapped to latent state 2 by the encoding process. This visualization demonstrates that when the transitions and rewards originate from a lower-dimensional manifold, the encoder contracts observations into equivalence classes, implying a small Lipschitz constant.

The desired state distribution (reward) is given by

$$P_R(X = n) = 0.99, \qquad P_R(X = 1) = 0.005, \qquad P_R(X \notin \{1, n\}) = \frac{0.005}{n - 2}.$$

### B.1.2. LATENT RIVERSWIM

We introduce the Latent RiverSwim environment that extends RiverSwim to a two-dimensional observation space

$$\mathcal{X} = \{1, \dots, n\}^2,$$

with action space

$$\mathcal{A} = \{(+1, 0), (-1, 0), (0, +1), (0, -1)\},$$

and a one-dimensional latent state space

$$\mathcal{S} = \{1, \dots, n\},$$

which defines the latent manifold. The latent encoding and transition dynamics are defined as

$$
\begin{aligned}
e_\alpha(i, j) &= \alpha i + (1 - \alpha)j, & (i, j) &\in \mathcal{X} & &\text{(state encoder)}, \\
k &= \lfloor e_\alpha(i, j) \rfloor & & & &\text{(latent state)}, \\
\bar{a} &= \text{sign}(e_\alpha(a_1, a_2)), & (a_1, a_2) &\in \mathcal{A} & &\text{(latent action)},
\end{aligned}
$$

for $\alpha \in (0, 1)$. The latent transition and reward dynamics follow those of RiverSwim with the identification $S \triangleq X$. The decoder maps a latent state $k$ back to the observation space by uniformly sampling one of the states consistent with the encoding:

$$P_D(X|S = k) = \mathcal{U}\big(\{X : \lfloor e_\alpha(i, j) \rfloor = k, \ (i, j) \in \mathcal{X}\}\big). \tag{10}$$

### B.1.3. TABULAR ALGORITHMS

We present the state of the art in tabular model-based reinforcement learning using a policy-iteration variant of posterior sampling for reinforcement learning (PSRL-PI), summarized in Algorithm 4. As a representative of state-of-the-art distributional reinforcement learning in the tabular setting, we consider Algorithm 5, which captures the key properties of implicit quantile networks. Our distributional active inference method can be implemented in tabular or discrete-action settings by extending the IQQL algorithm in Algorithm 5. We refer to the resulting method as *Distributional Active Inference (DAIF)*, which is presented in Algorithm 6. For details, see Appendix B.2.4.

---

**Algorithm 4** Tabular PSRL-PI

---

**Input:** $\pi(\cdot) := \mathcal{U}(\mathcal{A}), \gamma \in (0,1)$, Dirichlet concentration parameter $\alpha_{x,a} : \mathcal{X} \times \mathcal{A} \to \mathbb{R}_+^{|\mathcal{X}|}, \gamma \in (0,1)$
**while** True **do**
  $a = \pi(x)$
  $x', r := \texttt{env.step}(a)$
  $\alpha_{x,a}(x') := \alpha_{x,a}(x') + 1$
  $\widehat{P}_\alpha(\cdot|x,a) \sim \text{Dirichlet}(\alpha_{x,a}(\cdot)) \quad \forall x, a$
  **repeat**
    $V_\pi = \left(I - \gamma \widehat{P}_\alpha(x, \pi(x), \cdot)\right)^{-1} P_R$
    $\pi(x) := \arg\max_a P_R(x) + \gamma \sum_{x'} \widehat{P}_\alpha(x'|x,a) V_\pi(x') \quad \forall x \in X$
  **until** policy $\pi$ is stable
**end while**

---

**Algorithm 5** Tabular Implicit Quantile Q Learning (IQQL)

---

**Input:** $Q_\tau(\cdot, \cdot) := 0, \pi(\cdot) := \mathcal{U}(\mathcal{A}), D := \emptyset, \gamma \in (0,1)$
**while** True **do**
  $a := \pi(x)$
  $x', r := \texttt{env.step}(a)$
  $D := D \cup (x, a, r, x')$
  **repeat**
    $(\widetilde{x}, \widetilde{a}, \widetilde{r}, \widetilde{x}') \sim D, \qquad \tau, \tau' \sim \mathcal{U}(0,1)$
    $Q_\tau(\widetilde{x}, \widetilde{a}) := \arg\min_{Q'_\tau} \ell_\tau\left(\widetilde{r} + \gamma Q_{\tau'}(\widetilde{x}', \pi(\widetilde{x}')) - Q'_\tau(\widetilde{x}, \widetilde{a})\right)$       $\{\ell_\tau(u) \triangleq \frac{|u| + (2\tau - 1)u}{2}\}$
  **until** end of training epoch
  $\pi(x) := \arg\max_{a'} \mathbb{E}_{\tau'' \sim \mathcal{U}(0,1)}[Q_{\tau''}(x, a')] \quad \forall x \in X$
**end while**

---

### B.1.4. EXPERIMENTS

We evaluate the tabular methods described above on the RiverSwim and Latent RiverSwim environments using 50 independent repetitions. For RiverSwim, the agents interact with the environment for 5000 steps, while for Latent RiverSwim they interact for 10 000 steps. The first 10% of the interactions are performed using a random policy to initialize exploration. For the IQQL and DAIF models, we use neural networks to approximate the value functions. In RiverSwim, we employ a single linear layer, whereas in Latent RiverSwim we use a multilayer perceptron with ReLU activations and a single hidden layer of width 128. Figure 5 reports the results in terms of the frequency of visiting the most desired state within a 100-step window for varying horizon.

### B.2. Continuous control environments

#### B.2.1. EVOGYM

EvoGym (Bhatia et al., 2021) provides continuous control tasks for soft robots with deformable bodies, which introduce non-stationarity into the control dynamics and make the learning problem more challenging. As our focus is on the control optimization problem rather than robot morphology design, we do not design or evolve robot structures. Instead, we adopt the highest-reward robot morphologies provided in the EvoGym repository.[2] These morphologies were identified by Bhatia et al. (2021) through co-optimization of robot design and control, and represent structures that are well-suited to each task. By using these previously optimized designs, we isolate the control learning problem and ensure a fair comparison across algorithms without confounding effects from morphology variation. From the available environments, we choose seven tasks spanning different difficulty levels and task types to provide a comprehensive evaluation.

**Locomotion tasks.** We include four locomotion environments: (1) *Walker-v0* (easy), where the objective is to travel as far as possible in a single direction; (2) *UpStepper-v0* (medium), where the robot must climb stairs of varying heights; (3)

---

[2] https://huggingface.co/datasets/EvoGym/robots

---

**Algorithm 6** Tabular Distributional Active Inference (DAIF)

---

**Input:** $Q_\tau(\cdot,\cdot) := 0, \pi(\cdot) := \mathcal{U}(\mathcal{A}), D := \emptyset, \gamma \in (0,1)$
**while** True **do**
  $a := \pi(x)$
  $x', r := \texttt{env.step}(a)$
  $D := D \cup (x, a, r, x')$
  **repeat**
    $(\widetilde{x}, \widetilde{a}, \widetilde{r}, \widetilde{x}') \sim D, \qquad \tau, \tau' \sim \mathcal{U}(0,1)$
    $\mu, \alpha, \beta := Q_\tau(\widetilde{x}, \widetilde{a}), \qquad \mu' := Q_{\tau'}(\widetilde{x}', \pi(\widetilde{x}')), \qquad G := \widetilde{r} + \gamma\mu'$
    $Q_\tau(\widetilde{x}, \widetilde{a}) := \arg\max_{Q_\tau} \left\{ \log\tau(1-\tau) - \log\beta + \psi(\alpha) - \frac{\alpha}{2\beta}\left(|G - \mu| + (2\tau - 1)(G - \mu)\right) \right\}$
  **until** end of training epoch
  $\pi(x) := \arg\max_{a'} \mathbb{E}_{\tau'' \sim \mathcal{U}(0,1)}[Q_{\tau''}(x, a')] \quad \forall x \in X$
**end while**

---

*BidirectionalWalker-v0* (medium), where the target position changes dynamically during the episode, requiring the agent to learn locomotion in both directions; and (4) *Traverser-v0* (hard), where the robot must cross a pit without sinking.

**Object manipulation tasks.** We include three manipulation environments: (1) *Thrower-v0* (medium), where a box is placed on top of the robot and must be thrown over obstacles of varying sizes; (2) *Catcher-v0* (hard), where the robot must catch a rapidly falling and rotating box dropped from a height; and (3) *Lifter-v0* (hard), where the robot must lift a box out of a hole.

**Excluded environments.** From the original benchmark set, we exclude *Carrier-v0*, *BridgeWalker-v0*, *Climber-v0*, and *BeamSlider-v0*. We exclude *Carrier-v0* because it is classified as easy, and our evaluation prioritizes more challenging tasks while retaining one easy task as a baseline. We exclude *BridgeWalker-v0*, *Climber-v0*, and *BeamSlider-v0* because preliminary experiments showed minimal performance variance across algorithms in our initial seeds, limiting their discriminative value. Our final choice ensures balanced coverage across difficulty levels (one easy, three medium, three hard) and task types (four locomotion, three manipulation).

**Inclusion of non-benchmark environment.** We include *BidirectionalWalker-v0*, which is not part of the original benchmark set, because it introduces a qualitatively different locomotion challenge: the agent must learn to move in both directions in response to a dynamically changing goal, testing adaptability beyond standard unidirectional locomotion.

Figure 6 shows the learning curves across EvoGym environments, while Table 3 reports the numerical results in terms of AULC and final return.

### B.2.2. DEEPMIND CONTROL SUITE

DeepMind Control Suite (DMC) (Tassa et al., 2018) provides a diverse set of continuous control tasks built on the MuJoCo physics engine (Todorov et al., 2012) and is widely used as a benchmark for reinforcement learning algorithms. From the available environments, we choose eight tasks that span different robot morphologies and difficulty levels.

**Dog tasks.** The *dog* robot has the largest state and action space dimensionality in DMC. We include *dog-walk*, *dog-trot*, and *dog-run* to evaluate performance across increasing difficulty on this challenging morphology.

**Run tasks across robots.** We include *cheetah-run*, *walker-run*, *quadruped-run*, and *humanoid-run*. These robots range from planar (*cheetah*, *walker*) to 3D (*quadruped*, *humanoid*) and vary in state dimensionality and control complexity. We choose the *run* task as it is the most demanding locomotion variant for each robot.

Figure 7 presents the learning curves across DMC environments, while Table 4 reports the corresponding numerical results in terms of AULC and final return.

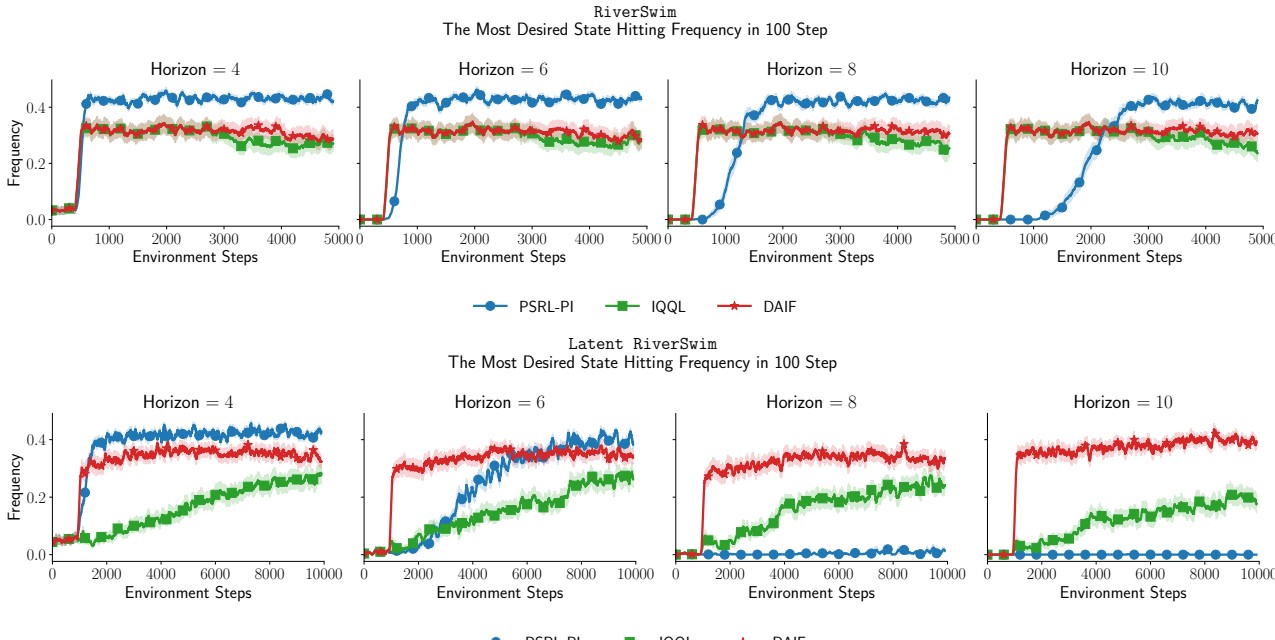

*Figure 5.* Top: RiverSwim. Bottom: Latent RiverSwim. Solid lines indicate the mean over 50 seeds, and shaded regions represent the standard error.

*Table 3.* Area Under the Learning Curve (AULC) and Final Return (mean $\pm$ standard deviation) averaged over 10 repetitions on the EvoGym environments. The highest mean values are highlighted in bold, and results within one standard deviation distance to the highest mean performance are underlined.

| Metric | Environment | Model | | | |
|---|---|---|---|---|---|
| | | DRND | DSAC | DTD3 | DAIF |
| AULC (↑) | Walker-V0 | $0.47 \pm_{0.54}$ | $5.54 \pm_{1.32}$ | $\mathbf{7.60} \pm_{\mathbf{1.24}}$ | $\underline{7.16} \pm_{1.47}$ |
| | Upstepper-V0 | $1.33 \pm_{0.17}$ | $0.88 \pm_{0.60}$ | $3.44 \pm_{0.92}$ | $\mathbf{5.56} \pm_{\mathbf{0.77}}$ |
| | Thrower-V0 | $0.86 \pm_{0.99}$ | $\underline{2.19} \pm_{0.48}$ | $\underline{2.71} \pm_{0.71}$ | $\mathbf{3.12} \pm_{\mathbf{1.17}}$ |
| | Bidirectionalwalker-V0 | $0.02 \pm_{0.01}$ | $4.68 \pm_{0.86}$ | $3.94 \pm_{2.09}$ | $\mathbf{7.21} \pm_{\mathbf{0.55}}$ |
| | Catcher-V0 | $-5.21 \pm_{0.06}$ | $-4.22 \pm_{0.23}$ | $-4.06 \pm_{0.41}$ | $\mathbf{-3.31} \pm_{\mathbf{0.39}}$ |
| | Lifter-V0 | $1.98 \pm_{0.26}$ | $1.88 \pm_{0.29}$ | $\underline{2.11} \pm_{0.28}$ | $\mathbf{2.27} \pm_{\mathbf{0.23}}$ |
| | Traverser-V0 | $-0.26 \pm_{0.18}$ | $0.92 \pm_{0.53}$ | $\underline{2.30} \pm_{0.25}$ | $\mathbf{2.43} \pm_{\mathbf{0.14}}$ |
| FINAL RETURN (↑) | Walker-V0 | $0.63 \pm_{0.75}$ | $7.78 \pm_{1.74}$ | $\mathbf{8.91} \pm_{\mathbf{0.86}}$ | $\underline{8.80} \pm_{1.55}$ |
| | Upstepper-V0 | $1.62 \pm_{0.06}$ | $0.97 \pm_{1.28}$ | $5.08 \pm_{2.07}$ | $\mathbf{8.35} \pm_{\mathbf{0.78}}$ |
| | Thrower-V0 | $1.17 \pm_{1.40}$ | $\underline{3.34} \pm_{0.80}$ | $\underline{3.38} \pm_{1.17}$ | $\mathbf{4.08} \pm_{\mathbf{1.64}}$ |
| | Bidirectionalwalker-V0 | $0.06 \pm_{0.15}$ | $6.11 \pm_{1.89}$ | $6.78 \pm_{2.41}$ | $\mathbf{8.79} \pm_{\mathbf{0.37}}$ |
| | Catcher-V0 | $-5.23 \pm_{0.20}$ | $-3.58 \pm_{0.39}$ | $-3.27 \pm_{0.93}$ | $\mathbf{-2.22} \pm_{\mathbf{1.01}}$ |
| | Lifter-V0 | $2.21 \pm_{0.72}$ | $\underline{2.34} \pm_{0.41}$ | $\underline{2.70} \pm_{1.06}$ | $\mathbf{2.74} \pm_{\mathbf{0.51}}$ |
| | Traverser-V0 | $0.16 \pm_{0.49}$ | $1.54 \pm_{1.14}$ | $\mathbf{2.85} \pm_{\mathbf{0.31}}$ | $\underline{2.62} \pm_{0.90}$ |

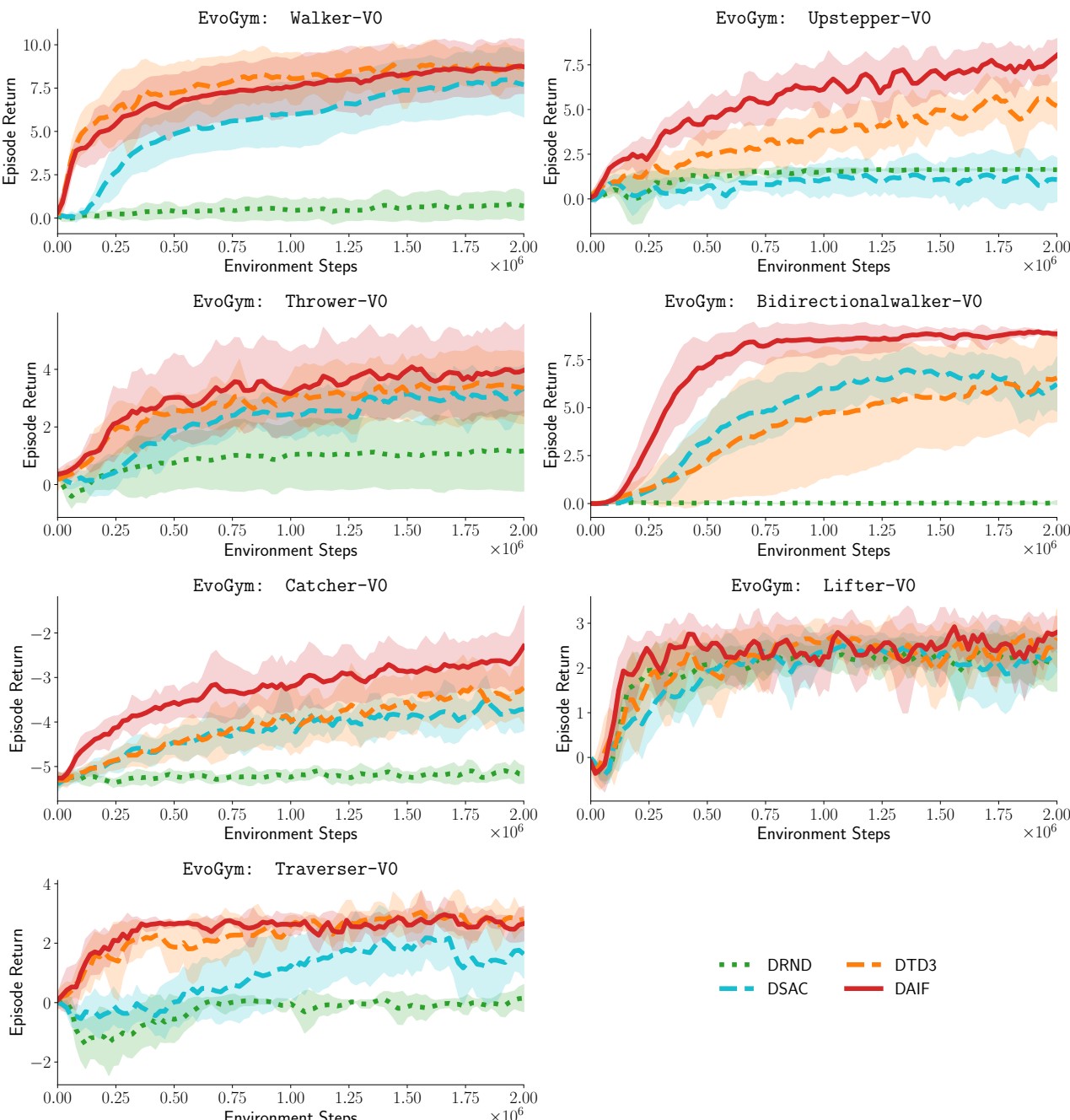

*Figure 6.* Learning curves for EvoGym environments.

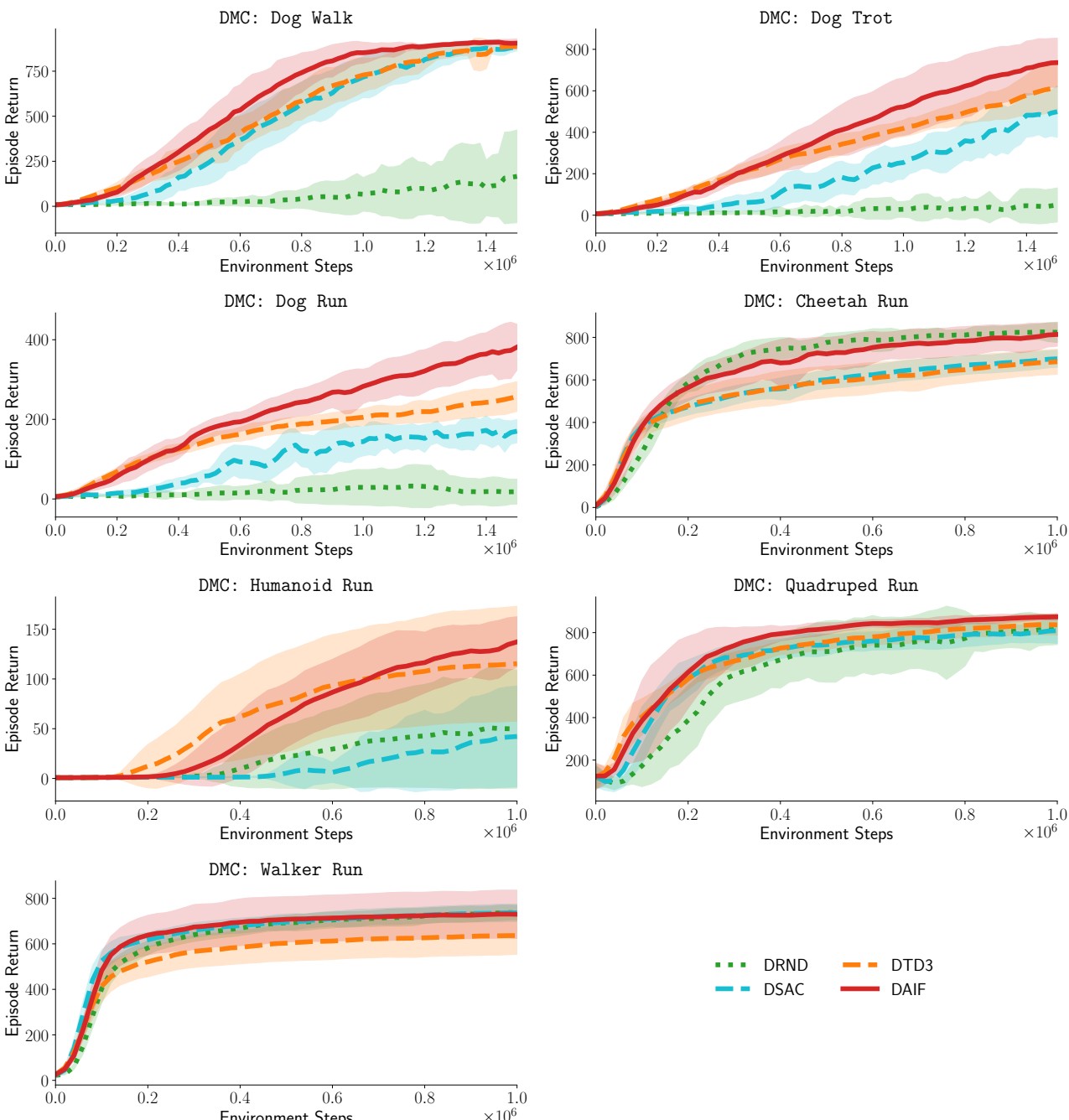

*Figure 7.* Learning curves for DeepMind Control suite environments.

*Table 4.* Area Under the Learning Curve (AULC) and Final Return (mean $\pm$ standard deviation) averaged over 10 repetitions on the DeepMind Control suite environments. The highest mean values are highlighted in bold, and results within one standard deviation distance to the highest mean performance are underlined.

| Metric | Environment | Model | | | |
|---|---|---|---|---|---|
| | | DRND | DSAC | DTD3 | DAIF |
| | Dog-Walk | $53.28 \pm_{74.94}$ | $468.31 \pm_{69.04}$ | $503.09 \pm_{53.94}$ | $\mathbf{575.84} \pm_{\mathbf{50.52}}$ |
| | Dog-Trot | $22.12 \pm_{30.23}$ | $189.95 \pm_{51.30}$ | $\underline{313.05} \pm_{28.88}$ | $\mathbf{368.90} \pm_{\mathbf{78.55}}$ |
| | Dog-Run | $17.92 \pm_{21.89}$ | $97.20 \pm_{19.02}$ | $162.40 \pm_{15.52}$ | $\mathbf{214.37} \pm_{\mathbf{31.33}}$ |
| AULC ($\uparrow$) | Cheetah-Run | $\mathbf{666.37} \pm_{\mathbf{30.09}}$ | $548.75 \pm_{38.19}$ | $540.36 \pm_{60.55}$ | $\underline{646.65} \pm_{56.03}$ |
| | Humanoid-Run | $22.02 \pm_{26.71}$ | $12.06 \pm_{15.63}$ | $\mathbf{66.22} \pm_{\mathbf{35.79}}$ | $\underline{61.56} \pm_{20.06}$ |
| | Quadruped-Run | $605.52 \pm_{101.59}$ | $658.64 \pm_{39.26}$ | $681.68 \pm_{28.00}$ | $\mathbf{720.89} \pm_{\mathbf{47.59}}$ |
| | Walker-Run | $\underline{615.66} \pm_{28.31}$ | $\underline{637.58} \pm_{31.87}$ | $546.43 \pm_{69.38}$ | $\mathbf{638.04} \pm_{\mathbf{88.82}}$ |
| | Dog-Walk | $171.67 \pm_{256.17}$ | $887.32 \pm_{21.58}$ | $887.64 \pm_{17.45}$ | $\mathbf{910.25} \pm_{\mathbf{20.23}}$ |
| | Dog-Trot | $53.55 \pm_{88.03}$ | $514.18 \pm_{121.13}$ | $612.97 \pm_{112.84}$ | $\mathbf{736.52} \pm_{\mathbf{117.58}}$ |
| | Dog-Run | $19.39 \pm_{31.78}$ | $184.70 \pm_{19.24}$ | $260.35 \pm_{35.76}$ | $\mathbf{382.27} \pm_{\mathbf{56.21}}$ |
| FINAL RETURN ($\uparrow$) | Cheetah-Run | $\mathbf{819.29} \pm_{\mathbf{58.72}}$ | $700.53 \pm_{39.77}$ | $685.59 \pm_{61.70}$ | $\underline{815.95} \pm_{52.73}$ |
| | Humanoid-Run | $51.92 \pm_{62.32}$ | $42.14 \pm_{51.26}$ | $\underline{115.84} \pm_{57.97}$ | $\mathbf{138.56} \pm_{\mathbf{25.29}}$ |
| | Quadruped-Run | $809.23 \pm_{81.57}$ | $809.65 \pm_{60.31}$ | $835.89 \pm_{46.84}$ | $\mathbf{873.91} \pm_{\mathbf{17.54}}$ |
| | Walker-Run | $\underline{736.29} \pm_{29.58}$ | $\mathbf{736.46} \pm_{\mathbf{38.15}}$ | $636.76 \pm_{81.69}$ | $\underline{730.13} \pm_{106.01}$ |

### B.2.3. DMC VISION.

The environments and tasks are identical to those in the state-based DMC setting; however, the agent learns directly from pixel observations rather than proprioceptive state inputs. Following the experimental setup of Yarats et al. (2022), we adopt their task difficulty categorization and hyperparameter configurations.

We evaluate on five tasks: *cheetah-run*, *quadruped-run*, *walker-run*, *reacher-hard*, and *finger-turn-hard*. The three *run* tasks test locomotion from pixels across different morphologies, while *reacher-hard* and *finger-turn-hard* provide non-locomotion tasks requiring precise control. We exclude *dog* tasks because Yarats et al. (2022) did not evaluate on this environment and no reference configurations are available. We also exclude *humanoid-run*, which is categorized as hard and requires $30\,000\,000$ frames, $10\times$ increase over the medium-difficulty tasks we consider.

We use DrQ-v2 as the backbone for our implementation, adopting DDPG-style exploration. The DTD3 variant corresponds to the distributional extension of DrQ-v2. We use the learning curves provided in the official repository[3]. Figure 8 presents the learning curves across DMC Vision environments, while Table 5 reports the corresponding numerical results in terms of AULC and final return.

### B.2.4. IMPLEMENTATION DETAILS

Given an observation tuple $(x, a, \tau)$, we define the input-dependent hyper-priors

$$\mu \triangleq \mu_\phi(x, a, \tau), \qquad \alpha \triangleq \alpha_\phi(x, a, \tau), \qquad \beta \triangleq \beta_\phi(x, a, \tau),$$

where $\mu_\phi$, $\alpha_\phi$, and $\beta_\phi$ are outputs of a neural network parameterized by $\phi$. We denote $\boldsymbol{m} \triangleq (\mu, \alpha, \beta)$. The corresponding generative model is defined as

$$\sigma \sim \mathcal{IG}(\alpha, \beta), \qquad G|\mu, \sigma, \tau \sim \mathcal{ALD}(\mu, \sigma, \tau),$$

where $\mathcal{ALD}$ denotes the asymmetric Laplace distribution with density

$$f(G|\mu, \sigma, \tau) = \frac{\tau(1-\tau)}{\sigma} \exp\left(-\ell_\tau\left(\frac{G-\mu}{\sigma}\right)\right),$$

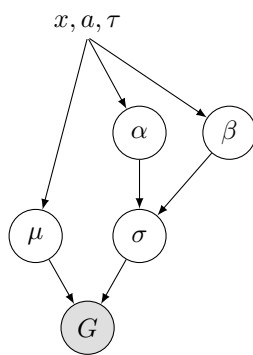

*Figure 9.* The plate diagram of the deep actor-critic implementation of DAIF. Each circle indicates a random variable. The observed random variable $G$ is shaded and the other ones are latent.

[3]https://github.com/facebookresearch/drqv2

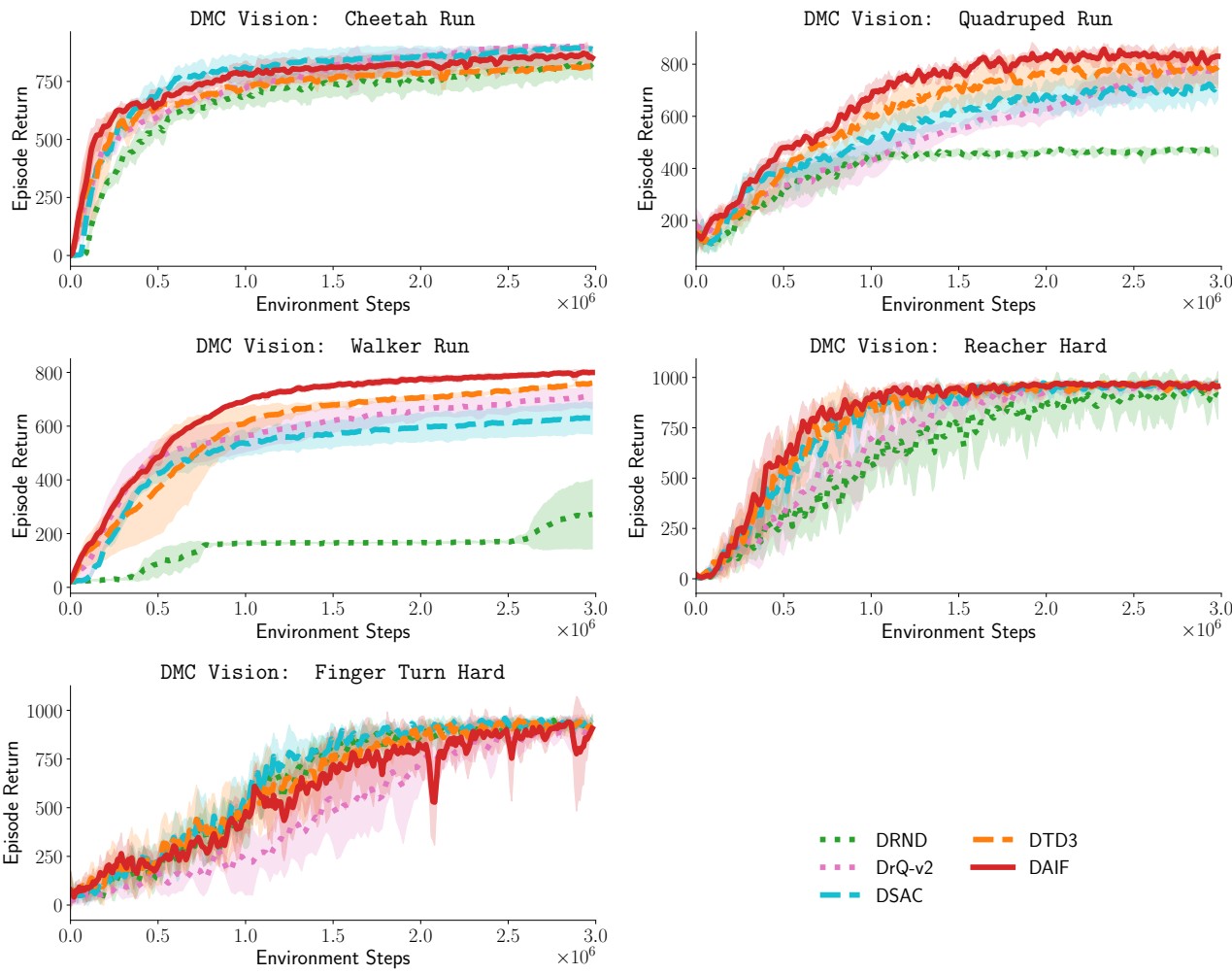

*Figure 8.* Learning curves for DeepMind Control suite vision environments.

*Table 5.* Area Under the Learning Curve (AULC) and Final Return (mean ± standard deviation) averaged over 5 repetitions on the DeepMind Control suite vision environments. The highest mean values are highlighted in bold, and results within one standard deviation distance to the highest mean performance are underlined.

| Metric | Environment | Model | | | | |
|---|---|---|---|---|---|---|
| | | DRND | DrQ-v2 | DSAC | DTD3 | DAIF |
| AULC (↑) | Cheetah-Run | $663.13 \pm_{38.98}$ | $\underline{743.02} \pm_{9.45}$ | $\mathbf{770.48} \pm_{\mathbf{33.96}}$ | $707.78 \pm_{17.05}$ | $\underline{756.45} \pm_{32.60}$ |
| | Quadruped-Run | $402.28 \pm_{10.70}$ | $526.64 \pm_{27.79}$ | $550.19 \pm_{41.06}$ | $613.70 \pm_{44.27}$ | $\mathbf{676.43} \pm_{\mathbf{18.12}}$ |
| | Walker-Run | $149.66 \pm_{10.64}$ | $565.91 \pm_{39.92}$ | $509.54 \pm_{42.86}$ | $587.99 \pm_{41.96}$ | $\mathbf{659.73} \pm_{\mathbf{6.49}}$ |
| | Reacher-Hard | $640.13 \pm_{96.90}$ | $706.24 \pm_{59.81}$ | $773.13 \pm_{12.60}$ | $\underline{783.05} \pm_{44.43}$ | $\mathbf{807.19} \pm_{\mathbf{27.02}}$ |
| | Finger-Turn Hard | $\underline{627.22} \pm_{23.88}$ | $481.89 \pm_{82.63}$ | $\mathbf{661.09} \pm_{\mathbf{35.50}}$ | $627.93 \pm_{59.60}$ | $580.21 \pm_{57.50}$ |
| FINAL RETURN (↑) | Cheetah-Run | $800.57 \pm_{61.43}$ | $\mathbf{894.25} \pm_{\mathbf{8.88}}$ | $880.97 \pm_{24.16}$ | $814.80 \pm_{8.87}$ | $859.80 \pm_{55.21}$ |
| | Quadruped-Run | $474.37 \pm_{8.94}$ | $761.76 \pm_{51.41}$ | $733.71 \pm_{55.48}$ | $\underline{796.24} \pm_{71.23}$ | $\mathbf{832.98} \pm_{\mathbf{54.93}}$ |
| | Walker-Run | $273.97 \pm_{130.92}$ | $699.36 \pm_{39.57}$ | $628.40 \pm_{59.10}$ | $761.66 \pm_{5.90}$ | $\mathbf{802.47} \pm_{\mathbf{5.40}}$ |
| | Reacher-Hard | $903.28 \pm_{132.70}$ | $\underline{969.82} \pm_{5.22}$ | $\mathbf{970.62} \pm_{\mathbf{13.02}}$ | $954.34 \pm_{36.76}$ | $954.84 \pm_{36.89}$ |
| | Finger-Turn Hard | $918.26 \pm_{85.21}$ | $941.10 \pm_{35.56}$ | $\mathbf{965.00} \pm_{\mathbf{4.97}}$ | $882.76 \pm_{92.06}$ | $920.58 \pm_{48.30}$ |

and the check function is defined as $\ell_\tau(u) = \dfrac{|u| + (2\tau - 1)u}{2}$. The corresponding log-likelihood is given by

$$\log f(G|\mu, \sigma, \tau) = \log \tau(1 - \tau) - \log \sigma - \frac{|G - \mu| + (2\tau - 1)(G - \mu)}{2\sigma}.$$

Rather than drawing reparameterized samples from the inverse-gamma distribution, we analytically marginalize over $\sigma$. Taking the expectation of the log-likelihood with respect to the inverse-gamma prior yields

$$\mathbb{E}_\sigma[\log f(G|\mu, \sigma, \tau)] = \log \tau(1 - \tau) - \mathbb{E}_\sigma[\log \sigma]$$
$$- \frac{1}{2}\left(|G - \mu| + (2\tau - 1)(G - \mu)\right)\mathbb{E}_\sigma\left[\frac{1}{\sigma}\right].$$

We first compute $\mathbb{E}_\sigma[\log \sigma]$. For $\sigma \sim \mathcal{IG}(\alpha, \beta)$,

$$\mathbb{E}_\sigma[\log \sigma] = \int_0^\infty \log \sigma \frac{\beta^\alpha}{\Gamma(\alpha)} \sigma^{-(\alpha+1)} \exp\left(-\frac{\beta}{\sigma}\right) d\sigma.$$

Applying the change of variables $a = \beta/\sigma$ yields

$$\mathbb{E}_\sigma[\log \sigma] = \frac{1}{\Gamma(\alpha)} \int_0^\infty (\log \beta - \log a) a^{\alpha-1} \exp(-a)\, da$$
$$= \log \beta - \psi(\alpha),$$

where $\psi(\cdot)$ denotes the digamma function. Next, we compute $\mathbb{E}_\sigma[1/\sigma]$. Using the same change of variables,

$$\mathbb{E}_\sigma\left[\frac{1}{\sigma}\right] = \frac{\beta^\alpha}{\Gamma(\alpha)} \int_0^\infty \sigma^{-(\alpha+2)} \exp\left(-\frac{\beta}{\sigma}\right) d\sigma = \frac{\alpha}{\beta},$$

where we used the identity $\Gamma(\alpha + 1) = \alpha\Gamma(\alpha)$. Combining the above results, we obtain

$$\mathbb{E}_\sigma[\log f(G|\mu, \sigma, \tau)] = \log \tau(1 - \tau) - \log \beta + \psi(\alpha) - \frac{\alpha}{2\beta}\left(|G - \mu| + (2\tau - 1)(G - \mu)\right). \tag{11}$$

We use the negative of this expected log-likelihood as the critic training objective. To mitigate numerical instabilities, we enforce $\alpha$ and $\beta$ to be larger than 10 by adding a constant offset of 10 to the corresponding network outputs. In addition, we adopt the regularization strategy proposed by Akgül et al. (2025) who placed weak hyperpriors

$$p(\mu) = \mathcal{N}(0, 1000^2), \qquad p(\alpha) = \mathcal{G}(10, 0.1), \qquad p(\beta) = \mathcal{G}(10, 0.1),$$

where the priors over $\alpha$ and $\beta$ are defined over the shifted parameters. We use this regularization with a regularization coefficient $\xi = 0.001$.

We adopt the design choices of Ma et al. (2025) combined with action-noise exploration and delayed policy updates (Fujimoto et al., 2018). The full update procedure is presented in Algorithm 7. The algorithm proceeds as follows: first, quantiles are randomly sampled and their midpoints are computed. Next, the target action is obtained from the lagged target actor with clipped Gaussian noise added. The parameters of the asymmetric Laplace distribution, $\mu$ and $\sigma$ are estimated for the current state-action pair and quantiles, where we model $\sigma \sim \mathcal{IG}(\alpha, \beta)$. Temporal difference errors are then computed for each critic using min-clipping. The critic loss is the negative log-likelihood weighted by pairwise quantile regression weights following Ma et al. (2025); Yang et al. (2019); specifically, line 23 weights each TD error by $(\tau_{i+1} - \tau_i)$, which corresponds to the quantile bin width and ensures that each quantile contributes proportionally to its probability mass under the target distribution. Regularization terms are computed using the priors, followed by parameter and Polyak updates for the critics. Every $d^{th}$ step (where $d = 2$), the policy is updated: quantiles and midpoints are recomputed, an action is sampled from the online actor, and the actor maximizes the estimated mean $\mu$ over the quantile distribution, followed by parameter and Polyak updates. For DMC vision control environments, we also apply the random augmentation method of Yarats et al. (2022) to our baselines.

---

**Algorithm 7** Distributional Active Inference update

---

1: **Parameters:** $N$: number of quantiles, $\gamma$: discount factor, $\iota$: Polyak averaging parameter, $\pi_\sigma$: policy exploration noise scale, $\pi_c$: policy noise clip bound, $d$: policy update delay interval, and $\xi$: regularization coefficient

2: **Networks:** $\pi_\theta$: policy network with parameters $\theta$, $\mu_{\phi_k}, \alpha_{\phi_k}, \beta_{\phi_k}$: critic networks ($k = 1, 2$) with parameters $\phi_k$, and $\bar\theta, \bar\phi_k$: target network parameters

3: **Input:** Transition $(x, a, r, x')$ from replay buffer

4: *— Critic Update —*

5: Sample and sort $\{\tau_i\}_{i=0}^N$, $\{\tau_j\}_{j=0}^N \sim \mathcal{U}(0, 1)$; compute midpoints $\hat\tau_i, \hat\tau_j$

6: $a' := \pi_{\bar\theta}(x') + \text{clip}(\epsilon, -\pi_c, \pi_c)$ where $\epsilon \sim \mathcal{N}(0, \pi_\sigma^2 I)$        ▷ *action-noise exploration*

7: Compute $\mu_j^k, \alpha_j^k, \beta_j^k$ from critic networks for $k \in \{1, 2\}, j \in \{0, \dots, N-1\}$

8: Compute TD errors: $u_{ij}^k := r + \gamma \min_{k'} \mu_{\bar\phi_{k'}}(x', a', \hat\tau_i) - \mu_j^k$ for all $i, j, k$        ▷ *Min-clipping*

9: *— Critic Loss (via Equation (11)) —*

10: **for** $k = 1, 2$ **do**

11:      $\mathcal{L}(\phi_k) := -\frac{1}{N} \sum_{i,j} (\tau_{i+1} - \tau_i) \left[ \log \hat\tau_j (1 - \hat\tau_j) - \log \beta_j^k + \psi(\alpha_j^k) - \frac{\alpha_j^k}{2\beta_j^k} \left( |u_{ij}^k| + (2\hat\tau_j - 1) u_{ij}^k \right) \right]$

12:      $\mathcal{L}(\phi_k) := \mathcal{L}(\phi_k) - \xi \frac{1}{N} \sum_j \left( \log p(\mu_j^k) + \log p(\alpha_j^k) + \log p(\beta_j^k) \right)$

13: **end for**

14: Update $\phi_k$ via $\nabla_{\phi_k} \mathcal{L}(\phi_k)$; $\bar\phi_k := \iota\phi_k + (1 - \iota)\bar\phi_k$ for $k = 1, 2$        ▷ *SGD + Polyak*

15: *— Policy Update —*

16: **if** $t \mod d = 0$ **then**

17:      Sample and sort $\{\tau_i\}_{i=0}^N \sim \mathcal{U}(0, 1)$; compute midpoints $\hat\tau_i$

18:      $\mathcal{L}(\theta) := -\frac{1}{2} \sum_{k,i} (\tau_{i+1} - \tau_i) \mu_{\phi_k}(x, \pi_\theta(x), \hat\tau_i)$

19:      Update $\theta$ via $\nabla_\theta \mathcal{L}(\theta)$; $\bar\theta := \iota\theta + (1 - \iota)\bar\theta$        ▷ *SGD + Polyak*

20: **end if**

---

### B.2.5. Training

**Baselines.** We include DRND (Yang et al., 2024) as a representative exploration-driven model-free reinforcement learning baseline. We also consider DSAC (Ma et al., 2025), the most recent state-of-the-art distributional actor–critic algorithm for continuous control. In addition, we introduce DTD3, a distributional extension of TD3 (Fujimoto et al., 2018), to directly assess whether DAIF improves over a distributional variant of TD3. For the DMC vision experiments, we include DrQ-v2 (Yarats et al., 2022), which achieves the highest reported scores among model-free methods on DMC vision environments. While model-based approaches such as DreamerV3 (Hafner et al., 2025) achieve strong performance, they require learning a world model, which is orthogonal to our focus on model-free policy optimization. Notably, at $1\,000\,000$ environment steps ($2\,000\,000$ frames), DAIF achieves competitive or superior performance compared to DreamerV3 on several tasks: DAIF outperforms DreamerV3 on *quadruped-run* (813 vs. 617), *walker-run* (780 vs. 684), and *reacher-hard* (950 vs. 862), while achieving comparable performance on *cheetah-run* (823 vs. 836) and *finger-turn-hard* (867 vs. 904). This demonstrates that our model-free approach can match or exceed the performance of state-of-the-art model-based methods without the additional complexity of world model learning.

**Critic architecture.** We adopt the critic architecture of Ma et al. (2025) for continuous control tasks, except for DMC Vision environments. The critic takes the state, action, and quantile fraction as inputs and outputs the parameters $(\mu, \alpha, \beta)$. The architecture follows a modular design with separate embeddings for the state–action pair and the quantile fraction, which are fused to produce the final quantile value estimates. Specifically, the state–action input is processed by a base network consisting of a linear layer with 256 hidden units, followed by layer normalization and a ReLU activation. The quantile fraction $\tau$ is embedded using a separate quantile network that maps a 128-dimensional quantile encoding to a 256-dimensional representation via a linear layer, layer normalization, and a sigmoid activation. The resulting embeddings are combined through element-wise interaction and passed to an output network comprising an additional hidden layer with 256 units, layer normalization, and ReLU activation, followed by a final linear layer that outputs scalar values for $\mu$, $\alpha$, and $\beta$.

*Table 6.* Area Under the Learning Curve (AULC) and Final Return (mean $\pm$ standard deviation) averaged over repetitions. The highest mean values are highlighted in bold, and results within one standard deviation of the best are underlined.

| Metric | Environment | Model | | | | | |
|---|---|---|---|---|---|---|---|
| | | DRND | DrQ-v2 | DSAC | DTD3 | DAIF | DAIF-SAC |
| AULC ($\uparrow$) | Catcher-V0 | $-5.21 \pm_{0.06}$ | — | $-4.22 \pm_{0.23}$ | $-4.06 \pm_{0.41}$ | $\mathbf{-3.31} \pm_{\mathbf{0.39}}$ | $-3.84 \pm_{0.23}$ |
| | Dog-Run | $17.92 \pm_{21.89}$ | — | $97.20 \pm_{19.02}$ | $162.40 \pm_{15.52}$ | $\mathbf{214.37} \pm_{\mathbf{31.33}}$ | $138.08 \pm_{27.49}$ |
| | Quadruped-Run | $402.28 \pm_{10.70}$ | $526.64 \pm_{27.79}$ | $550.19 \pm_{41.06}$ | $613.70 \pm_{44.27}$ | $\mathbf{676.43} \pm_{\mathbf{18.12}}$ | $512.26 \pm_{78.71}$ |
| FINAL RETURN ($\uparrow$) | Catcher-V0 | $-5.23 \pm_{0.20}$ | — | $-3.58 \pm_{0.39}$ | $-3.27 \pm_{0.93}$ | $\mathbf{-2.22} \pm_{\mathbf{1.01}}$ | $-3.53 \pm_{0.57}$ |
| | Dog-Run | $19.39 \pm_{31.78}$ | — | $184.70 \pm_{19.24}$ | $260.35 \pm_{35.76}$ | $\mathbf{382.27} \pm_{\mathbf{56.21}}$ | $299.70 \pm_{50.61}$ |
| | Quadruped-Run | $474.37 \pm_{8.94}$ | $761.76 \pm_{51.41}$ | $733.71 \pm_{55.48}$ | $\underline{796.24 \pm_{71.23}}$ | $\mathbf{832.98} \pm_{\mathbf{54.93}}$ | $702.88 \pm_{92.04}$ |

For DMC Vision control tasks, in addition to the critic architecture described above, we employ the convolutional encoder of Yarats et al. (2022) to process high-dimensional observations. The encoder consists of four convolutional layers with 32 channels each. The first layer uses a stride of 2, while the remaining layers use stride 1. Each convolution is followed by a ReLU activation, and the resulting feature maps are flattened to obtain a compact latent representation. After the encoder, we use a Linear layer that maps the output of the encoder to a feature dimension of 50, followed by layer normalization and $\tanh$ activation for both actor and critic.

**Actor architecture.** We parametrize the actor by a multilayer perceptron that maps the state representation to a continuous action. The network consists of two hidden layers with 256 units each, each followed by a ReLU activation. A final linear layer outputs the action vector, which is subsequently passed through a $\tanh$ activation to enforce bounded actions within the valid action range. This architecture provides sufficient expressive capacity for modeling complex continuous policies while remaining computationally efficient and stable in practice. For DMC Vision control tasks, the same actor architecture is used on top of the encoder, and a DDPG-style exploration scheme is applied following Yarats et al. (2022).

**Optimization.** We use a learning rate of $3 \times 10^{-4}$ for both the actor and the critic, except for DMC Vision control tasks, where a learning rate of $1 \times 10^{-4}$ is used for the actor, critic, and encoder. We employ a replay buffer of size $1\,000\,000$ and a batch size of 256, with $10\,000$ environment steps for warm-up. The discount factor is set to 0.99, and Polyak averaging with a coefficient of 0.005 is used for target network updates. We use two critics and a single actor, with target networks maintained for both. Policy smoothing is applied using Gaussian noise with standard deviation 0.1, target policy noise with standard deviation 0.2, and target noise clipping of 0.5, together with a policy update delay of 2. We randomly sample 8 quantiles for value estimation, target computation, and actor training. The regularization coefficient is set to $\xi = 0.001$. For DMC Vision tasks, we use a frame stack of 3 and an action repeat of 2, and all other hyperparameters follow Yarats et al. (2022)[4].

**Environment interactions.** We interact with the environments for $1\,000\,000$ steps for DMC control tasks, except for dog environments, where we use $1\,500\,000$ interaction steps. For EvoGym environments, we perform $2\,000\,000$ interaction steps, while for DMC Vision tasks we train for $3\,000\,000$ environment frames. All experiments are repeated 10 times, except for DMC Vision tasks, where we use 5 repetitions.

**Ablation study on DAIF with SAC.** We also investigate the performance of DAIF with SAC version (DAIF-SAC) on three representative environments chosen from each suite. As shown in Table 6, DAIF-SAC performs well across all three environments, consistently outperforming its base algorithm DSAC on Catcher-V0 and Dog-Run, while achieving comparable performance on Quadruped-Run. These results demonstrate that DAIF can be effectively applied to distributional versions of actor-critic algorithms beyond TD3. It is worth noting that in the DMC Vision tasks, DTD3 and DAIF are based on DrQ-v2, which employs DDPG as its backbone, further highlighting the flexibility of our approach across different actor-critic frameworks.

**Wall-clock time comparison.** We measure wall-clock training time for $10\,000$ environment interaction steps on the *Dog-Run* task from the DMC suite, using a system equipped with a GeForce RTX 4090 GPU, an Intel Core i7-14700K CPU (5.6 GHz), and 96 GB memory. DTD3 requires 59.69 seconds, DAIF 66.81 seconds, DSAC 75.05 seconds, and DRND

---

[4] https://github.com/facebookresearch/drqv2/tree/c0c650b76c6e5d22a7eb5f2edffd1440fe94f8ef

81.69 seconds. Relative to its baseline DTD3, DAIF incurs an overhead of approximately 12% in wall-clock time. These results indicate that the performance improvements achieved by DAIF come at only a modest additional computational cost over its underlying actor–critic architecture.

## C. Extended Related Works

### C.1. Distributional Reinforcement Learning

Distributional RL explicitly models the full return distribution rather than only its expectation. While it was initially introduced for value-based algorithms (Bellemare et al., 2017; Dabney et al., 2018b;a; Yang et al., 2019), subsequent work extended these ideas to actor–critic architectures, particularly in continuous control settings (Barth-Maron et al., 2018). Various distributional actor–critic variants have been proposed to address specific limitations, including training instability arising from categorical critics (Nam et al., 2021; Singh et al., 2022), overestimation bias (Kuznetsov et al., 2020; Li et al., 2024; Döring et al., 2025), and risk-sensitive decision making (Ma et al., 2025), with several approaches focusing on risk-aware locomotion control (Shi et al., 2024; Schneider et al., 2024; Zhang et al., 2025). Our work generalizes the distributional reinforcement learning framework by working with push-forward mappings of trajectory measures, where the new push-forward RL formulation extends the distributional RL. This generalization allows us to integrate the active inference framework into distributional RL, which in turn lets us develop model-free RL algorithms for active inference. The term *push-forward* also appears in Bai et al. (2025), but there it refers to transport-map parameterizations of action/return distributions from base noise, whereas our push-forward RL pushes forward trajectory measures within the Bellman theory. Therefore the overlap is primarily terminological. In our experiments, we compare our method against distributional RL baselines such as the distributional Soft Actor–Critic (DSAC) (Ma et al., 2025) and Implicit Quantile Q-Learning) (Dabney et al., 2018a). To our knowledge, our method is the first to integrate active inference into the distributional reinforcement learning framework.

### C.2. Directed Exploration in Reinforcement Learning

Directed exploration explicitly guides reinforcement learning agents toward uncertain or informative states, proving essential in non-stationary environments with changing dynamics and sparse-reward settings lacking frequent feedback. While directed exploration has been extensively studied in the context of classical reinforcement learning algorithms (Houthooft et al., 2016; Bellemare et al., 2016; Pathak et al., 2017; Ostrovski et al., 2017; Burda et al., 2019; Ciosek et al., 2019; Yang et al., 2024), distributional reinforcement learning also provides a natural foundation for exploration. In particular, distributional losses have been shown to induce intrinsic exploration through uncertainty-aware regularization (Sun et al., 2025). Beyond this implicit effect, several works explicitly design exploration strategies within the distributional reinforcement learning framework. Tang & Agrawal (2018) exploit epistemic uncertainty in return distributions for posterior sampling-like exploration; Mavrin et al. (2019) introduce Decaying Left Truncated Variance to derive an optimistic bonus from upper-tail quantiles with decaying optimism; Zhou et al. (2021) propose Distributional Prediction Error, using Wasserstein distance between quantile distributions to construct exploration bonuses; Oh et al. (2022) schedule risk levels from risk-seeking to risk-averse to balance exploration; and Cho et al. (2023) randomize risk criteria through distributional perturbations to avoid biased optimism. While these methods explicitly design exploration bonuses, the exploration term in our distributional actor–critic objective arises naturally from variational free-energy minimization. For directed exploration baselines, we also consider DSAC, which has been shown to exhibit intrinsic exploration effects (Sun et al., 2025). In addition, we include DRND (Yang et al., 2024), a recent state-of-the-art exploration method for actor–critic algorithms.

### C.3. Active Inference and Reinforcement Learning

Active inference has increasingly been examined in relation to reinforcement learning as an alternative that unifies perception, action, and learning by minimizing expected free energy (EFE) without explicit rewards or value functions (Friston et al., 2009; 2015; Ueltzhöffer, 2018; Sajid et al., 2021a; Çatal et al., 2019; Tschantz et al., 2020b; Fountas et al., 2020; Millidge et al., 2020; Millidge, 2020; Hafner et al., 2020b). A growing body of empirical work shows that active inference can solve standard reinforcement learning benchmarks, often achieving performance comparable to classical reinforcement learning algorithms (Cullen et al., 2018; van der Himst & Lanillos, 2020; Tschantz et al., 2020a; Marković et al., 2021; Paul et al., 2021). Despite these advantages, particularly the intrinsic balancing of goal-seeking and information gain through epistemic value, most practical AIF agents remain *model-based*, relying on learned generative/world models, belief (variational) inference, and often explicit planning or lookahead to evaluate and optimize EFE (Lanillos et al., 2021; Tschantz et al.,

2020a; Fountas et al., 2020; Schneider et al., 2022; Sajid et al., 2021b). This reliance contributes to the computational burden of multi-step EFE estimation and has historically kept many scalable demonstrations either in discrete settings or in continuous-control implementations with relatively short effective planning horizons (Lanillos et al., 2021; Da Costa et al., 2023). Closest in spirit to our work, Malekzadeh & Plataniotis (2024) derive Bellman-style recursions for EFE and instantiate actor–critic updates; however, their formulation operates in belief space and still depends on learned belief representations and a learned world model for objective construction/evaluation, whereas our approach targets the model-free setting.

