# OpenReview forum: "Distributional Active Inference"
_ICML.cc/2026/Conference — ICML 2026 regular_

### Official Review · Reviewer_am4Y · 2026-03-10

**Soundness:** 3
**Presentation:** 2
**Significance:** 3
**Originality:** 3
**Overall Recommendation:** 4
**Confidence:** 2

**Summary:**

The work proposes a new theoretical framework of push-forward RL that connects the distribution of returns with the transition function, connecting model-based RL and distribution RL algorithms. This framework is step-by-step derived from first principles in Bayesian and active causal inference and reveals a new perspective and connection on distribution RL and model-based RL algorithms. From these theoretical findings, a new new RL algorithm in distribution active inference (DAIF) is proposed that implements distributional RL optimisation without the need to learn a world model. In empirical experiments across continuous control tasks, DAIF is shown to outperform standard distributional RL baselines and is shown to scale better to long-horizon tasks compared to standard distribution RL and model-based baselines.

**Compliance With Llm Reviewing Policy:**

Affirmed.

**Final Justification:**

As expressed in my rebuttal response and review, my confidence for this assessment is fairly low since I'm not familiar with related theory and the literature but I lean towards acceptance as indicated by my score.

**Key Questions For Authors:**

1. In the introduction, the work states that "distributional RL provides a computationally efficient alternative to model-based RL". However, I would have seen distributional RL (modelling the distribution of returns rather than point-estimates) and model-based RL (explicitly learning a model of transition/ reward functions of the environment) as orthogonal approaches that can and have been combined, rather than alternatives to each other. Why do you state these as alternatives to each other?
2. Lines 200ff (right column) state that a given fixed point is unique but it is unclear why it is clear for this fixed point to be unique. From where do you follow that the fixed point must be unique?
3. Lines 310ff (left column) state that "Using the expectation of the identified push-forward distributions in the policy improvement step, which is the Bayes-optimal way to make predictions with it, ...". Why is this prediction guaranteed to be Bayes-optimal?
4. Lines 369ff (left column) state that "the other steps of our uncertainty quantification pipeline already induce the necessary randomness for exploration". I do not find it obvious how the uncertainty quantification induces exactly the randomness needed for exploration. Would the authors be able to elaborate on why the induced randomness achieves the necessary exploration?

**Limitations:**

yes.

**Strengths And Weaknesses:**

To provide context for the following review, I'd like to state that while I am very familiar with empirical RL research and, to some extent, with distributional and model-based RL approaches, but I am less familiar with some of the mathematical concepts used within the theoretical contributions of this work. I found several parts of this work, in particular Section 3, quite difficult to parse. I will provide concrete pointers to challenges I encountered in the weaknesses section below which hopefully can help strengthen this work's presentation.

## Strengths
I appreciate that this work builds bridges between different communities, revealing, to the best of my knowledge, new connections between distribution RL, model-based RL, and active inference. Such connections can be the foundation of follow-up work and the empirically demonstrated promise of the proposed distributional active inference (DAIF) already establishes that the theoretical contributions are valuable and can lead to significant advancements of the field.

This combination of theoretical contributions and a comprehensive empirical evaluation is rare and represent strong contributions to the field.

To the best of my ability, the theoretical contributions appear sound but have several instances of unclear or overloaded notation that make it difficult to parse (see weakness below). The methodology of the empirical evaluation, including the selection of baselines and environments, are representative of challenging continuous control tasks.

## Weaknesses
I believe the main weakness of this work lies within its presentation. Frequent re-use and occasional unclear notation, unclear statements, significant parts of the work being deferred to the appendix, and a lack of contextualization of why these contributions are meaningful make it difficult to parse and appreciate this work.

Overall, these weaknesses make me wonder whether this work would be better suited for a different venue, possibly a journal that allows for a more comprehensive submission. This would allow to properly introduce all notation, provide context to help the reader understand made contributions and allow to fit all made contributions within the paper, rather than having to defer critical parts to the Appendix.

Below, I will give concrete pointers for each of these challenges.

**Unclear statements:**
1. In the introduction, the work states that "distributional RL provides a computationally efficient alternative to model-based RL". However, I would have seen distributional RL (modelling the distribution of returns rather than point-estimates) and model-based RL (explicitly learning a model of transition/ reward functions of the environment) as orthogonal approaches that can and have been combined, rather than alternatives to each other. Why do you state these as alternatives to each other?
2. Lines 200ff (right column) state that a given fixed point is unique but it is unclear why it is clear for this fixed point to be unique. From where do you follow that the fixed point must be unique?
3. Lines 310ff (left column) state that "Using the expectation of the identified push-forward distributions in the policy improvement step, which is the Bayes-optimal way to make predictions with it, ...". Why is this prediction guaranteed to be Bayes-optimal?
4. Lines 369ff (left column) state that "the other steps of our uncertainty quantification pipeline already induce the necessary randomness for exploration". I do not find it obvious how the uncertainty quantification induces exactly the randomness needed for exploration. Would the authors be able to elaborate on why the induced randomness achieves the necessary exploration?

**Unclear or overloaded notation:**

5. In the problem statement, the world model $P_W(X, S)$ is defined but the functional notation is not given. What does $P_W$ map to? Is it $P_W: \mathcal{X} \times \mathcal{S} \mapsto \mathcal{X}$? or perhaps mapping to the latent state space $\mathcal{S}$?
6. In Section 2, actions and perception are defined with $Y$ and $S$, while in the problem statement above, actions were denoted with $A$ and $S$ denoted latent auxiliary variables. These inconsistencies make reading the work harder than it needs to be.
7. In line 104, the world model $P_W$ is re-defined, now also conditioned on actions $Y$. Lines 125ff again re-define it with additional conditioning and modelled variables. The same is done with $P_Q$ which is first defined inline 55f (right column) but then keeps being redefined with different conditioning and variables, e.g. in lines 134ff (left column).
8. The first definition of $f_{P_0}(Y := y)$ uses the notation of $Y \sim \delta_y$ which is only defined in Appendix Table 2. But even with this Appendix table, this statement is unclear to me since it only defines $\delta_x(X)$ with a set $X$ but there appears to be no set given to $\delta_y$?
9. In the definition of $P_W$ in lines 125ff, the last term denotes $P_0(\pi)$ which I believe is a mistake. Given the following text, I believe this should write $P_A(\pi)$ for the policy distribution. Also, $P_E$ is stated to be "an encoder that maps the sensory stimulus space to the latent perception space" but in notation it is written as $P_E(S | X)$ and $X$ has previously been defined as the environment state which seem distinct from sensory stimulus (which sounds more like perception)?
10. Lines 155 (right column), you write that "for high-dimensional (vector) state spaces, $|\mathcal{S}|$ can grow rapidly ..." but $\mathcal{S}$ denotes the latent space of perception/ auxiliary variables. I would expected it to read $|\mathcal{X}|$ for the environment state space instead?
11. Line 272 (right column) refers to $\mathcal{W}_2$ but I am unable to find a definition of this notation. What does this refer to?

**Core contributions in Appendix:**

I believe that the Appendix of this work goes significantly beyond material that can be described as supplementary but rather contains core contributions and context necessary to parse parts of this work. Providing proofs for theoretical contributions and additional experimental details for reproducibility and potential supplementary experiments are all reasonable, but deferring definitions of algorithmic contributions, related work, and key context on empirical experiments are all components that I would expect to find in the main paper. Below, I list the material from the Appendix in this work that I would have expected to find in the main paper.

12. Notation used throughout the paper should be clearly defined in the main paper. Table 2 can still be useful to have a single point of reference but none of the definitions should only appear in this Appendix table.
13. To be able to understand the empirical experiments in the RiverSwim environment, I would expect at least high-level descriptions of these tasks. What are their key parameters, how is the horizon varied and how exactly does the latent and standard version differ. This is only presented in Appendix B.1.2. Similarly, the choice of tasks and baselines of the deep RL evaluation are explained in Appendix B.2.
14. I'd argue that in its current form, the main paper does not present the deep RL DAIF algorithm as a core contribution. Instead, the algorithm, including its objectives, are only defined in Appendix B.2.4. which goes significantly beyond stating implementation details.
15. The related work section of Appendix C goes beyond what I would expect to see in the main paper, but I would expect large parts of this being contextualized in the main paper. I believe Section 5 is insufficient to contextualize this work within the wider literature. Adding some of these discussions would significantly help the reader to appreciate the contributions of this work.

---

> ### Author Rebuttal · Authors · 2026-03-28
>
> **On distributional RL as an alternative to model-based RL**
>
> We agree that the two are orthogonal algorithmically. However, they are often used for a shared purpose: finding the return-maximizing policy with minimal interactions. A distributional approach can find the fixed point with a *set* of transition kernels (paragraph at Line 190, right column), whereas model-based RL must recover the *true* kernel, a harder problem. The statement was made in the context of active inference: prior AIF+RL work requires model learning for EFE, and distributional RL captures the return distribution without an explicit forward model. We will rewrite the sentence as: "Distributional RL captures the full return distribution, encoding information about future state transitions that would otherwise require an explicit forward model. This makes distributional RL a natural vehicle for incorporating AIF without the cost of learning an explicit transition model."
>
> **On uniqueness of the fixed point**
>
> Uniqueness follows directly from Banach's fixed-point theorem applied to the contraction in Lemma 3.1. Since $\gamma < 1$, the distributional Bellman operator is a strict contraction on the complete metric space, guaranteeing exactly one fixed point. We will make this reasoning explicit.
>
> **On Bayes-optimal prediction**
>
> Policy improvement depends on the expectation of the learned return distribution, whose mean coincides with the value function. Taking the expectation over the encoding measure recovers the standard action-value function by the tower property of conditional expectation, see Bellemare et al. 2023 MIT Press. We will sharpen this statement and add the reference.
>
> **On exploration from uncertainty quantification**
>
> Epistemic uncertainty is captured by the Inverse-Gamma posterior on sigma_tau, inducing posterior uncertainty over the return distribution's spread at each quantile. For each $(x,a,\tau)$, the encoder outputs $(\mu, \alpha, \beta)$, where $\alpha, \beta$ parameterize an Inverse-Gamma on the ALD scale sigma. The ratio $\beta/\alpha$ reflects expected scale, propagating directly into critic updates (Alg. 7). Early in training and in rarely visited states, the broad posterior yields high-variance value estimates; acting on these naturally explores uncertain regions, analogous to Thompson sampling (Osband et al., 2013). Complementarily, Sun et al. (2025) show distributional losses intrinsically promote exploration via uncertainty-aware regularization, aligning with our finding that explicit entropy bonuses were redundant.
>
> **On notation concerns**
>
> We thank the reviewer for the detailed list and address each point:
> - $P_W(X,S)$ maps to $\mathcal P_{\mathcal X \times \mathcal S}$, the set of probability measures on the product space.
> - Section 2 derives AIF in its general form where $Y$ represents actions and $S$ perception; we then switch to control-specific variables. We will add a clarifying sentence at the transition.
> - The progressive redefinitions of $P_W$ and $P_Q$ reflect a constructive buildup; each step refines rather than contradicts the previous one. We will flag this pattern explicitly.
> - $\text{do}(Y \sim \delta_y)$ is equivalent to $\text{do}(Y=y)$, written stochastically for notational consistency with the remaining interventions. $\delta_y$ is the Dirac measure (Table 2).
> - $P_0$ vs $P_A$ in Line 125: indeed a typo. Regarding $P_E$: we will unify terminology by consistently using "observation" for $\mathcal X$ throughout, avoiding the ambiguity between "sensory stimulus" and "environment state."
> - $|\mathcal S|$ vs $|\mathcal X|$ was a typo.
> - $\mathcal W_2$ is the p-Wasserstein distance for $p=2$ (Table 2). We will define in the main text.
>
> **On core contributions in the appendix**
>
> We will expand the notation paragraph to include all definitions currently only in Table 2, and introduce $\mathcal W_p$ and the do-operator in the main text. We added high-level descriptions of the RiverSwim environments and baselines to Section 6. Regarding Alg 7: our main contribution is conceptual: making AIF applicable to model-free, infinite-horizon settings. Alg 3 is the meta-algorithm; Algs 6 and 7 are specific instantiations. Using the extra camera-ready page, we will move the closed-form critic loss (Eq. 10) into the main text, reference Algorithm 7, and summarize key design choices in Section 4. We will expand Section 5 into a structured related work section with: (1) AIF and RL, positioning DAIF against model-based AIF; (2) Distributional RL, distinguishing our push-forwards from prior distributional methods; and (3) Exploration in distributional RL, contextualizing our implicit exploration against explicit bonus methods. App. C will be retained for broader coverage.
>
> **On the conceptual figure**
>
> We will add a new Figure 1 to the Introduction; see our response to Reviewer vp6r for details. Preliminary version: https://anonymous.4open.science/r/DAIF/DAIF_conceptual_overview.pdf.

---

> > ### Author Rebuttal · Reviewer_am4Y · 2026-04-01
> >
> > I thank the authors for their clarifications. I believe that inclusion of the mentioned context and information in the main paper, using the extra page for a camera ready version of the paper, will significantly improve the clarity of the work. of address most of my concerns.
> >
> > As expressed in my review, my assessment is fairly low confidence given my lack of familiarity with the presented theory, but I lean towards acceptance and will raise my score to "Weak accept".

---

### Official Review · Reviewer_vp6r · 2026-03-12

**Soundness:** 3
**Presentation:** 2
**Significance:** 3
**Originality:** 3
**Overall Recommendation:** 5
**Confidence:** 3

**Summary:**

The paper proposes an approach, at the intersection between active inference and distributional reinforcement learning (RL), for solving complex control problems, called Distributional Active Inference. They first provide an alternative formulation of active inference based on distributional RL, then build a new theoretical framework with push-forward operators for return distributions, and finally present a new active inference method based on this theory.

**Compliance With Llm Reviewing Policy:**

Affirmed.

**Final Justification:**

Technically sound and well written manuscript. The proposed method is validated against state-of-the-art algorithms and non-trivial control tasks. Pseudo algorithms and implementation details are provided.

**Key Questions For Authors:**

I do not have any further questions.

**Limitations:**

Yes

**Strengths And Weaknesses:**

The presented paper is technically sound and well written. The authors clearly state the current limitations of RL and of state-of-the-art implementations of active inference, and propose a rigorous and detailed formulation of their method. The authors validate the proposed approach with several state-of-the-art RL algorithms and non-trivial control tasks, providing extensive results and discussion about related works on the Supplementary Materials. Pseudo algorithms and implementation details are also provided to allow replication of the results.

The proposed paper have several strengths. First, active inference is a new brain-based and computationally efficient theory, hence advancing this subfield could provide valid alternatives -- compared to standard optimal control -- to deal with tasks that biological organisms tackle with small samples and few computational burden; some of these advantages are shown in the results section. Second, adapting active inference to distributional RL allowed to develop model-free algorithms for the considered framework; in contrast, many state-of-the-art implementations of active inference suffer in scenarios with highly complex models, limiting scalability. While the audience may be modest for the time being, I believe the paper is of high utility and that advancing in the direction proposed by the authors will bring many benefits in the future compared to more traditional solutions.
This work provides a better understanding on the links between active inference and RL, which are often overlooked. For this, the authors build a novel computational framework called push-forward RL that allows them to build the proposed method.

One weakness is that the presentation is math-heavy and it may be challenging to follow, so the authors could provide more intuition (especially in chapter 3) to improve the flow of the paper, possibly adding a graphical representation of the theoretical framework introduced.

Also, the authors could use different tick symbols in Figure 2 so that the different algorithms are better distinguishable for color-blind people or in b/w.

---

> ### Author Rebuttal · Authors · 2026-03-28
>
> **On the conceptual figure**
>
> We will add a figure to the introduction section that illustrates our key message and give a reference in the fifth paragraph of the introduction. The figure will highlight that our DAIF equips the role of the free energy concept of active inference with two features: (i) model-free infinite-horizon planning, (ii) incorporating the complete free energy distribution into policy search instead of only its expectation. We share a preliminary version of this figure in the following link: https://anonymous.4open.science/r/DAIF/DAIF_conceptual_overview.pdf.
>
> **On improving intuition and flow**
>
> We have taken several steps to improve accessibility. Beyond the new conceptual figure, we moved part of the technical discussion into the appendix and added clarifications in response to specific notation and clarity concerns raised by Reviewers qCCK and am4Y (see our responses to those reviewers for details). In particular, we expanded the notation paragraph, added clarifying sentences at key transition points (e.g., when switching from general AIF variables to the control-specific setting), and flagged the progressive construction pattern of $P_W$. Using the extra page granted for the camera-ready version, we will move the closed-form critic loss (Eq. 10) into the main text, explicitly reference Algorithm 7, and add a high-level description of the RiverSwim environments and baselines to Section 6.
>
> **On color-blind accessibility**
>
> Thank you for this suggestion. We will update all figures with distinct tick symbols and line styles so that algorithms are distinguishable in both color and black-and-white printing.

---

> > ### Author Rebuttal · Reviewer_vp6r · 2026-04-03
> >
> > Thank you for addressing my questions.

---

### Official Review · Reviewer_mnum · 2026-03-12

**Soundness:** 3
**Presentation:** 3
**Significance:** 3
**Originality:** 3
**Overall Recommendation:** 4
**Confidence:** 2

**Summary:**

The paper proposes a measure-theoretic framework, push-forward reinforcement learning, that reinterprets distributional RL as push-forwards of Markov process measures through trajectory functionals. Building on this view, the authors present a derivation of active inference (AIF) using variational and causal inference tools and claim a simplification of the AIF objective under an intervention on observations by a desired state distribution. They then introduce Distributional Active Inference (DAIF), a model-free algorithm that performs quantile-style temporal-difference learning in a latent, amortized probabilistic embedding, and report improved performance over strong distributional actor-critic baselines across tabular and continuous-control benchmarks.

**Compliance With Llm Reviewing Policy:**

Affirmed.

**Key Questions For Authors:**

1. In environments where state abstractions are not advantageous, is the 12% computational overhead of DAIF justified compared to vanilla distributional RL?
2. Why was action-noise exploration empirically superior to the maximum-entropy policy search prescribed by the AIF ELBO?

**Limitations:**

Yes

**Strengths And Weaknesses:**

Strengths:
1. Recasts distributional RL rigorously via push-forwards of trajectory measures, making explicit the role of Markov kernels and return functionals and connecting model-based and model-free updates.
2. Introduces Theorem 3.5 tying representation bottlenecks (through Lipschitz encoder/decoder) to the contraction modulus of the distributional Bellman operator, providing a principled argument for abstraction aiding convergence.
3. Attempts a unification between AIF and distributional RL by operationalizing “predictive coding” ideas without an explicit forward model.

Weakness:
1. Implementing DAIF results in approximately a 12% increase in wall-clock computation time compared to standard distributional actor-critic methods.
2. The expected gains in sample complexity depend on the capacity of function approximators to accurately represent the necessary distributions and the level of compression permitted by the underlying transition dynamics

---

> ### Author Rebuttal · Authors · 2026-03-28
>
> **On the computational overhead**
>
> In environments without beneficial abstractions, DAIF matches (but does not degrade below) vanilla distributional RL, as shown in the RiverSwim left panel of Figure 2 and in environments like Walker-V0 (Table 3). The 12% overhead is relative to DTD3 and modest in absolute terms (66.8s vs. 59.7s per 10K steps). Importantly, DAIF has *lower* overhead than other vanilla distributional RL methods: DSAC requires 75.05s (+26%) and DRND requires 81.69s (+37%). Since DAIF never hurts and often substantially helps, the modest overhead is justified as a default choice.
>
> **On the dependence on function approximator capacity**
>
> This dependency is shared with every deep RL method that uses function approximation and is not specific to DAIF. What DAIF adds in Theorem 3.5 is that it makes the dependence on $L_E \cdot L_D$ explicit rather than leaving it implicit in the approximation error.
>
> **On action-noise exploration vs. maximum-entropy policy search**
>
> The epistemic uncertainty is captured by the Inverse-Gamma posterior on $\sigma_\tau$, which creates posterior uncertainty over the scale of the return distribution at each quantile level. For each $(x,a,\tau)$, the encoder outputs $(\mu, \alpha, \beta)$, where alpha and beta parameterize an Inverse-Gamma on the ALD scale sigma. The ratio beta/alpha reflects the expected scale, i.e., the uncertainty of the return distribution, which propagates into the critic updates (Alg. 7). Early in training and in rarely visited states, the posterior over sigma is broad, yielding high variance in value estimates. When the policy selects actions based on these noisy estimates, it naturally explores uncertain regions of the state space,  analogous to Thompson sampling (Osband et al., 2013), where exploration arises from acting on posterior samples rather than point estimates. Complementarily, Sun et al. (2025) show that distributional losses intrinsically promote exploration via uncertainty-aware regularization, providing a complementary perspective on our finding that explicit entropy bonuses were redundant.
>
> **On the conceptual figure**
>
> We will add a conceptual figure to the Introduction (see our response to Reviewer vp6r for details and a preliminary version). The figure highlights that DAIF equips the free energy concept of active inference with two features: (i) model-free infinite-horizon planning and (ii) incorporating the complete free energy distribution into policy search instead of only its expectation.

---

> > ### Author Rebuttal · Reviewer_mnum · 2026-04-02
> >
> > Thank you for addressing my and other reviewers' comments.

---

### Official Review · Reviewer_qCCK · 2026-03-13

**Soundness:** 3
**Presentation:** 2
**Significance:** 4
**Originality:** 4
**Overall Recommendation:** 6
**Confidence:** 3

**Summary:**

As far as the reviewer understands, this paper consists of the following contributions:

- Theoretical results on distributional bellman operators when pushed to a lower dimensional manifold. This insight is used in an algorithm that can take advantage of both distributional-RL and state abstraction.
- An interpretation of this algorithm as transition-free/ model-free active inference.
- Empirical demonstration of performance over full state distributional alternatives.

**Compliance With Llm Reviewing Policy:**

Affirmed.

**Final Justification:**

Ultimately, my high score reflects the novelty of the approach combining the strengths of model based and distributional RL, the technical justification of the algorithm and the empirical success against what appear to be strong baselines.

The concerns I raised centred on the space spent at the beginning of the paper relating distirbutional reinforcement learning to active inference, which does not seem to be necessary to motivate the state-abtracted distributional algorithm they present. As a consequence, the more relevant technical details are very condensed and this reduces the clarity of the paper.  The rebuttals given by the authors did not convince me this tradeoff was worthwhile.

**Key Questions For Authors:**

- Would the algorithm work with a deterministic encoder, i.e. parameterising $g_{\tau}$, $\sigma_{\tau}$, etc directly?
- Is there some notion of epistemic uncertainty I am missing?
- How is the size of the state-space found, clearly if the bottleneck is too small the distinctions in the return distribution can dissappear — relatedly, where does this tradeoff appear in the theoretical section?
- Why is it natural to expect the decoder lipschitz constant $L_{D}$ to be small?

**Limitations:**

The authors point out the limitation of finite-sample gaurantees. A good adddition would be the limitations as a description of active inference.

**Strengths And Weaknesses:**

**Strengths**:

- The empirical results show excellent performance and compared to strong baselines on good range of challenging tasks.
- The integration of state abstraction into distributional RL is novel and as far as the reviewer is aware and seems likely to generate further research.
- The theoretical analysis of how encoder-decoders affect the rate of contraction of the associated bellman operator appears rigorous and likely to be useful for future research.

**Weaknesses**:

The paper is currently very hard to digest. In particular:
- The prioritisation of an active inference interpretation, do-operation etc. — which seems tangential to understanding the final algorithm — could be relegated to the appendix.
- Some of the mathematics is overly technical in the body of the paper. For example the existence of the measure, and the detailed definitions of push-foward could be stated intuitively and the details put in the appendix.
- The paper would greatly benefit from an explanatory schematic showing how the encoder-decoder and distributional objects interact with each other.
- The section about $E_{\phi}$ and the role of $g_{\tau}$ and $\mu_{\tau}$, etc. was especially hard to digest.
- Could benefit from moving Algorithm 7 in the appendix into the main text, or at least referencing in the main text.

The role of active inference, the argument that this should be seen as active inference — as opposed to control via approximate inference is weak.

- There is no EFE-like aspect — arguably core to an active inference algorithm. (Though the reviewer agrees this just acts as an heuristic).
In particular seemingly no meaningful no operational use of epistemic uncertainty — maybe it implicitly appears in the taking the expectation of  $E_{\phi}$.
- It is not clear why there is pressure for the encoder to reduce the size of the state-space —- is this just enforced by a hyperparameter —- if so there is the, not small, problem of tuning this.
- The narrative emphasises that a natural advantage of AIF is to cope with computational limits —- which is then operationalised as operating in an abstracted state space —- however this doesn’t seem related to active inference at all.

---

> ### Author Rebuttal · Authors · 2026-03-28
>
> **On expected free energy and epistemic uncertainty**
>
> Section 2 derives the complete active inference formula in Eq. 3. The EFE corresponds to the second and third RHS terms, arising from the KL divergence between posterior and prior policy distributions. The posterior is updated for future interactions based on the world model, while the prior is a Boltzmann distribution with the negative ELBO as energy function (right column, Lines 69–72), computed w.r.t. an alternative decomposition of the joint over $X$ and $Y$ where the reward function serves as the desired state distribution—fully in line with core AIF principles. The latent return distribution in DAIF directly inherits the second term of Eq. 3, leaving the third term optional.
>
> We presented the derivation from an ELBO perspective for the ML readership, rather than the variational free energy view common in neuroscience-oriented AIF. We will clarify after Line 73 (right column) that the exponent's interior is the negative of the expected free energy as known in the AIF literature.
>
> Regarding epistemic uncertainty: it is captured by the Inverse-Gamma posterior over $\sigma_\tau$, inducing posterior uncertainty over the return distribution's spread at each quantile level. For each $(x,a,\tau)$ the encoder outputs $(\mu,\alpha,\beta)$, where $\alpha,\beta$ parameterize an Inverse-Gamma on the ALD scale $\sigma$. The ratio $\beta/\alpha$ reflects the expected scale, propagating directly into critic updates (Alg. 7). Early in training and in rarely visited states, the broad posterior over $\sigma$ yields high variance in value estimates; acting on these noisy estimates naturally explores uncertain regions—analogous to Thompson sampling (Osband et al., 2013). Complementarily, Sun et al. (2025) show that distributional losses intrinsically promote exploration via uncertainty-aware regularization, aligning with our finding that explicit entropy bonuses were redundant.
>
> **On the size of the state space**
>
> Reducing the state dimension creates equivalence classes whose elements are equivalent w.r.t. further processing. When the latent state has equal or more dimensions, raw stimuli become more beneficial.
>
> The latent state space is purely a theoretical construction for DAIF, introducing no hyperparameters. Eq. 5 shows such a space always exists, but the learner operates on its push-forward measure in return-distribution space, latent representations are never explicitly constructed. $E_\phi$ models epistemic uncertainty from state-abstraction ambiguity on the return distribution (an infinite-horizon EFE distribution).
>
> **On latent representations and active inference**
>
> Latent representations' instrumental value is not prescribed by core AIF; AIF is a meta-theory modeling neural processes at all levels, where each step maps the previous output into a latent space. Our hypothesis is that the practical benefit for control lies in equivalence classes from state abstractions, following Abel et al. (2016, 2018). Theorem 3.5 supports this: state abstractions create equivalence classes via encoding, and this operation has a small Lipschitz $L_E$ constant **(not $L_D$)** as classes grow, making the encoder less sensitive to input perturbations. Theorem 3.5 shows that such an encoder within an AIF pipeline reduces the contraction modulus, accelerating convergence to the optimal policy.
>
> **On presentation density**
>
> We sympathize with this concern. The density inevitably follows from bridging AIF and distributional RL, two technical fields with distinct concepts and notations. Our goal is to present rigorous AIF derivations and measure-theoretic machinery establishing a shared vocabulary. This vocabulary is novel: the do-calculus component shows the EFE has a simpler equivalent than widely known, enabling Eq. 3; the measure-theoretic concepts connect kernel machinery to Theorem 3.5's convergence analysis, our main theoretical contribution of potentially independent interest. All intermediate steps are deferred to Appendices A.4 and A.5.
>
> Upon acceptance, we will use the extra camera-ready page to improve accessibility.
> We'll include a conceptual figure into the main text. See our response to vp6r.
>
> **On a deterministic encoder**
>
> A deterministic encoder could yield a working distributional RL algorithm. Theorem 3.5's contraction improvement holds for a deterministic measurable map $S:\mathcal{X}\to\mathcal{S}$, so the convergence guarantee is retained.
>
> The stochastic encoder specifically provides the AIF connection: the variational posterior $q(s\mid x)$ is required for the ELBO objective, for propagating epistemic uncertainty over $\sigma_\tau$ into exploration, and for the non-degenerate composite return (Eq. 5). A deterministic variant reduces to distributional RL with learned Lipschitz representations, reasonable, but forfeiting AIF-specific benefits (variational objective, intrinsic exploration, free-energy interpretation) that are this work's core contribution.

---

> > ### Author Rebuttal · Reviewer_qCCK · 2026-04-03
> >
> > Thank you for the detailed response. I will keep my high score as my concerns are more at a semantic level.
> >
> > My concerns about the active inference interpretation remain. There is a history in the literature of overuse of the term active inference, and I feel the paper would be stronger with a smaller emphasis on this aspect.  However my earlier worry about no-EFE was too strong.
> >
> > The authors helped clarify that the uncertainty over quantiles induces uncertainty _about the reward distribution_ given a latent encoding, however would they agree there is no uncertainty about the observation to latent mapping, or transition model (ultimately stemming from the form of the variational model $P_{Q}$ just above Eq.3).
> >
> > More fundamentally you explicitly state “prescribes a Dyna-style model based approach” so why not start here. The relationship to the active inference under specfic assumptions is interesting but seemingly not fundamental, nor providing any further clarity into the structure of the algorithm. As a result the paper is trying to fill two roles, and ends up reducing the clarity.

---

> > > ### Author Response · Authors · 2026-04-03
> > >
> > > We are enthusiastic to discuss the implications of our work further. We fully agree that active inference is an overloaded concept. This is precisely why we took the original paper that established the mathematical foundations of the field [1]. All the material presented in our paper related to active inference stems only from analytical steps taken over the version presented in this paper. The variational posterior $P_Q$ is a design choice both in Bayesian machine learning and active inference. In the absence of a theoretical obligation, we follow the probabilistic state space models best practices and choose it as in Line 135, i.e. $P_Q := P_M P_E P_A$. This choice eliminates the related KL term wrt the prior dynamics. However, the induced uncertainty appears in the second term of Eq. 3 while the expectation of the desired state distribution (reward) is being taken. Notably, while vanilla active inference collapses this uncertainty to a point estimate by taking its expectation, we capture the total uncertainty of state abstractions as well as the randomness in the policy, state transitions, and rewards. This uncertainty is modeled by the implicit quantile network that quantifies its push-forward effect on the return distribution. The multi-stage effect of the uncertainty on the latent push-forward trajectories can be seen in the hierarchical structure presented in Figure 8 of the appendix.
> > >
> > >
> > > We are thankful to the reviewer's efforts to enhance the target audience of our work. We do think that the contribution of our work is not limited to the novel algorithm and its demonstrated practical benefits. In particular, Section 2 contributes to active inference theory by developing its complete machinery for the first time sticking exclusively to statistical inference elements. Our use of the "do" operator clarifies some new relationships among building blocks, for example ending up with the surprising outcome in Eq. 1. The material developed in Section 2 also has the descriptive value demonstrated in Section 4. This section proves formally that our algorithm inherits the benefits of an active inference agent. This proof explains the outcome presented in the synthetic experiment given in Figure 2.
> > >
> > >
> > > All the above being said, if the reviewer has any specific suggestions about how to improve the presentation further, we are very happy to accommodate.
> > >
> > > [1] Friston et al., Active Inference: A Process Theory, Neural Computation, 29, 1-49, 2017

---

### Decision · Program_Chairs · 2026-04-30

**Decision:**

Accept (regular)

**Comment:**

This paper is technically sound, reasonably well-written, novel, and useful to those in the ICML community. Reviewers generally agreed on the novelty of the contribution, and several reviewers pointed out presentation issues that make the paper denser and less-intuitive than they would have preferred.